# Hemodynamic differences between women and men with elevated blood pressure in China: A non-invasive assessment of 45,082 adults using impedance cardiography

**César Caraballo**[1,2], **Shiwani Mahajan**[1,2], **Jianlei Gu**[3,4], **Yuan Lu**[1], **Erica S. Spatz**[1,2], **Rachel P. Dreyer**[5], **MaoZhen Zhang**[6,7], **NingLing Sun**[8], **Yihong Ren**[9], **Xin Zheng**[10], **Hongyu Zhao**[3,11], **Hui Lu**[3,12], **Zheng J. Ma**[3,11,13], **Harlan M. Krumholz**[1,2,14]*

**1** Center for Outcomes Research and Evaluation, Yale New Haven Hospital, New Haven, Connecticut, United States of America, **2** Department of Internal Medicine, Section of Cardiovascular Medicine, Yale School of Medicine, Yale University, New Haven, Connecticut, United States of America, **3** SJTU-Yale Joint Center for Biostatistics, School of Life Science and Biotechnology, Shanghai Jiao Tong University, Shanghai, China, **4** Shanghai Engineering Research Center for Big Data in Pediatric Precision Medicine, Shanghai, China, **5** Department of Emergency Medicine, Yale School of Medicine, New Haven, CT, United States of America, **6** iKang Healthcare Group, Inc., Shanghai, China, **7** Department of Cardiology, Xinhua Hospital Affiliated with Shanghai Jiao Tong University, Shanghai, China, **8** Department of Hypertension at Heart Center, Peking University People's Hospital, Beijing, China, **9** The First Medical Center of Chinese PLA General Hospital, Beijing, China, **10** National Clinical Research Center of Cardiovascular Diseases, State Key Laboratory of Cardiovascular Disease, Fuwai Hospital, National Center for Cardiovascular Diseases, Chinese Academy of Medical Sciences and Peking Union Medical College, Beijing, China, **11** Department of Biostatistics, School of Public Health, Yale University, New Haven, Connecticut, United States of America, **12** Center for Biomedical Informatics, Shanghai Children's Hospital, Shanghai, China, **13** Beijing Li-Heng Medical Technologies, Ltd, Beijing, China, **14** Department of Health Policy and Management, Yale School of Public Health, New Haven, Connecticut, United States of America

* harlan.krumholz@yale.edu

**Data Availability Statement:** iKang Health Group provided the de-identified data used in this study to SJTU-Yale Joint Center for Biostatistics in

## Abstract

### Background

Whether there are sex differences in hemodynamic profiles among people with elevated blood pressure is not well understood and could guide personalization of treatment.

### Methods and results

We described the clinical and hemodynamic characteristics of adults with elevated blood pressure in China using impedance cardiography. We included 45,082 individuals with elevated blood pressure (defined as systolic blood pressure of ≥130 mmHg or a diastolic blood pressure of ≥80 mmHg), of which 35.2% were women. Overall, women had a higher mean systolic blood pressure than men (139.0 [±15.7] mmHg vs 136.8 [±13.8] mmHg, P<0.001), but a lower mean diastolic blood pressure (82.6 [±9.0] mmHg vs 85.6 [±8.9] mmHg, P<0.001). After adjusting for age, region, and body mass index, women <50 years old had lower systemic vascular resistance index (beta-coefficient [β] -31.7; 95% CI: -51.2, -12.2) and higher cardiac index (β 0.07; 95% CI: 0.04, 0.09) than men of their same age group, whereas among those ≥50 years old women had higher systemic vascular resistance index

Shanghai, China. This is where all data are stored and all analyses were performed. There are recent strict legal restrictions in publicly sharing data from China outside of its borders (see Personal Information Protection Law, November 2021). Requests can be sent to iKang Health at zhaohui. wang@ikang.com for consideration to share data in a legally compliant manner. The code used to analyze the data is publicly available at https:// www.doi.org/10.5281/zenodo.5931975.

**Funding:** The authors received no specific funding for this work.

**Competing interests:** Yuan Lu is supported by the National Heart, Lung, and Blood Institute (K12HL138037) and the Yale Center for Implementation Science. She was a recipient of a research agreement, through Yale University, from the Shenzhen Center for Health Information for work to advance intelligent disease prevention and health promotion. Erica S. Spatz receives support from the Food and Drug Administration to support projects within the Yale-Mayo Clinic Center of Excellence in Regulatory Science and Innovation (CERSI); the National Institute on Minority Health and Health Disparities (U54MD010711-01) to study precision-based approaches to diagnosing and preventing hypertension; and the National Institute of Biomedical Imaging and Bioengineering (R01EB028106-01) to study a cuff-less blood pressure device. Xin Zheng is supported by the CAMS Innovation Fund for Medical Science (2016-I2M-1-006), the National Key Research and Development Program (2016YFE0103800) from the Ministry of Science and Technology of China. Hui Lu is supported by the National Key R&D Program of China (2018YFC0910500) and Neil Shen's SJTU Medical Research Fund. Zheng J. Ma is affiliated with Beijing Li-Heng Medical Technologies, Ltd, which designed the ICG device used in this study. Harlan Krumholz works under contract with the Centers for Medicare & Medicaid Services to support quality measurement programs; was a recipient of a research grant, through Yale, from Medtronic and the U.S. Food and Drug Administration to develop methods for post-market surveillance of medical devices; was a recipient of a research grant with Medtronic and is the recipient of a research grant from Johnson & Johnson, through Yale University, to support clinical trial data sharing; was a recipient of a research agreement, through Yale University, from the Shenzhen Center for Health Information for work to advance intelligent disease prevention and health promotion; collaborates with the National Center for Cardiovascular Diseases in Beijing; receives payment from the Arnold & Porter Law

(β 120.4; 95% CI: 102.4, 138.5) but lower cardiac index (β -0.15; 95% CI: -0.16, -0.13). Results were consistent with a propensity score matching sensitivity analysis, although the magnitude of the SVRI difference was lower and non-significant. However, there was substantial overlap between women and men in the distribution plots of these variables, with overlapping areas ranging from 78% to 88%.

## Conclusions

Our findings indicate that there are sex differences in hypertension phenotype, but that sex alone is insufficient to infer an individual's profile.

## Introduction

Mean arterial pressure is determined by cardiac output (CO) and systemic vascular resistance (SVR), and there are important sex-specific differences in its regulation and the risk of developing hypertension [1–4]. Recent studies [5–7] have shown that, on average, women with hypertension have a higher SVR and a lower CO when compared with men. Such observations suggest that sex could serve as a proxy for the underlying hemodynamic phenotype among individuals with elevated blood pressure. Considering that tailoring antihypertensive treatment based on individuals' hemodynamic profile may be associated with better blood pressure control [8–11], these sex differences on hemodynamic phenotypes could help to identify more personalized therapeutic approaches. However, current hypertension guidelines have no sex-specific recommendations on therapy–other than in pregnancy–due to a lack of evidence of benefit from sex-stratified therapies [12,13].

Most of the studies addressing these underlying hemodynamic sex differences have had small samples or focused exclusively on younger individuals [5,6], which limits the generalizability of their findings to a broader population that includes the elderly, among whom the hypertension burden is greater [14–16]. Particularly, after menopause, women experience a sharp increase in hypertension prevalence [17–19], eventually surpassing aged-matched men [20]. Thus, assessing the sex-specific differences in hemodynamic profiles across age groups can provide a better understanding on whether there are substantial hemodynamic differences by sex that could be used to guide therapy.

Accordingly, we used data from tens of thousands of individuals with elevated blood pressure from an outpatient setting in China to evaluate the overall patterns of sex differences in hemodynamic variables and to determine how these sex hemodynamic differences may vary with age. We also aimed to evaluate the distribution of these variables among women and men, and to what extent they overlap by sex. Furthermore, we stratified our analysis at the mean age of menopause in China because of its association with hemodynamic changes. Results from this study can advance our understanding of the association of sex with hemodynamic patterns in people with hypertension and suggest if sex could be used to guide therapy.

## Methods

### Data source

iKang Health Group provided the de-identified data used in this study to SJTU-Yale Joint Center for Biostatistics in Shanghai, China. This is where all data are stored and all analyses were performed. There are recent strict legal restrictions in publicly sharing data from China

Firm for work related to the Sanofi clopidogrel litigation, from the Ben C. Martin Law Firm for work related to the Cook IVC filter litigation, and from the Siegfried and Jensen Law Firm for work related to Vioxx litigation; chairs a Cardiac Scientific Advisory Board for UnitedHealth; was a participant/ participant representative of the IBM Watson Health Life Sciences Board; is a member of the Advisory Board for Element Science, the Advisory Board for Facebook, and the Physician Advisory Board for Aetna; and is the founder of HugoHealth, a personal health information platform, and co-founder of Refactor Health, an enterprise healthcare AI-augmented data management company. The other co-authors report no potential competing interests. This does not alter our adherence to PLOS ONE policies on sharing data and materials.

outside of its borders (see Personal Information Protection Law, November 2021). Requests can be sent to iKang Health at zhaohui.wang@ikang.com for consideration to share data in a legally compliant manner. The code used to analyze the data is publicly available at https://www.doi.org/10.5281/zenodo.5931975.

## Study population

Between January 2012 and October 2018, 116,851 individuals (65,172 men and 51,679 women) underwent an impedance cardiography (ICG) test offered as part of the employee routine annual physical examination at 51 sites of iKang Health Checkup Centers throughout China. We excluded those younger than 20 years and those older than 80 years (n = 839). Then, we excluded 1,814 individuals with outlier values of weight, height, blood pressure, heart rate, stroke volume, and baseline thoracic impedance (**Fig 1**). Of the 114,198 remaining individuals, we included 45,082 with elevated blood pressure, which was defined as a systolic blood pressure (SBP) of ≥130 mmHg or a diastolic blood pressure (DBP) of ≥80 mmHg, consistent with the 2017 American College of Cardiology/American Heart Association hypertension guidelines [12].

## Data collection

At health centers, nurses collected information on the patient's age, sex, geographical region, weight, height, SBP, and DBP. Weight was measured using a calibrated and standardized scale, rounded to the nearest 0.1 kg. Height was measured to the nearest 0.1 cm using a portable stadiometer (Omron HNJ-318; Omron Corporation, Kyoto, Japan) with patients standing without shoes and heels against the wall. Body mass index (BMI) was calculated as weight in kilograms divided by the square of height in meters. After 5 minutes of resting in a seated position, blood pressure was measured once using an automated monitor (Omron HBP-9020; Omron Corporation, Kyoto, Japan) on the right arm.

Patients were then requested to lay supine and, after 3 minutes in this position, all hemodynamic parameters were measured using ICG. The ICG method that has been validated against invasive techniques for estimation of stroke volume and CO in both stable and high-risk populations [21–23], and has been shown to be a highly reproducible technique [24]. By applying a constant, low amplitude, high-frequency, alternating electrical current to the thorax, ICG device measures the corresponding voltage to detect beat-to-beat changes in thoracic electrical resistance, known as impedance, and with it stroke volume is estimated [25,26]. Then, using heart rate, mean arterial blood pressure, and BMI, other hemodynamic parameters are calculated, including CO, cardiac index (CI), SVR, and systemic vascular resistance index (SVRI) [27]. The ICG device used (CHM T3002/P3005, designed by Beijing Li-Heng Medical Technologies, Ltd, manufactured by Shandong Baolihao Medical Appliances, Ltd.) was developed based on improved hardware and advanced digital filtering algorithms [28], and has been validated versus both invasive thermodilution and non-invasive echocardiography in different settings [29–31].

## Variable definitions

We described demographic characteristics and hemodynamic parameters of blood pressure in women and men overall and by age. Considering that the mean age of natural menopause in China is reported as approximately 50 years of age [32–34], we stratified our study population as <50 years old and ≥50 years old. We used the World Health Organization recommended cutoff values for BMI classification in Asian populations, defining underweight as <18.5 kg/m$^2$, normal weight from 18.5 kg/m$^2$ to <23 kg/m$^2$, overweight from 23 kg/m$^2$ to <27.5 kg/m$^2$,

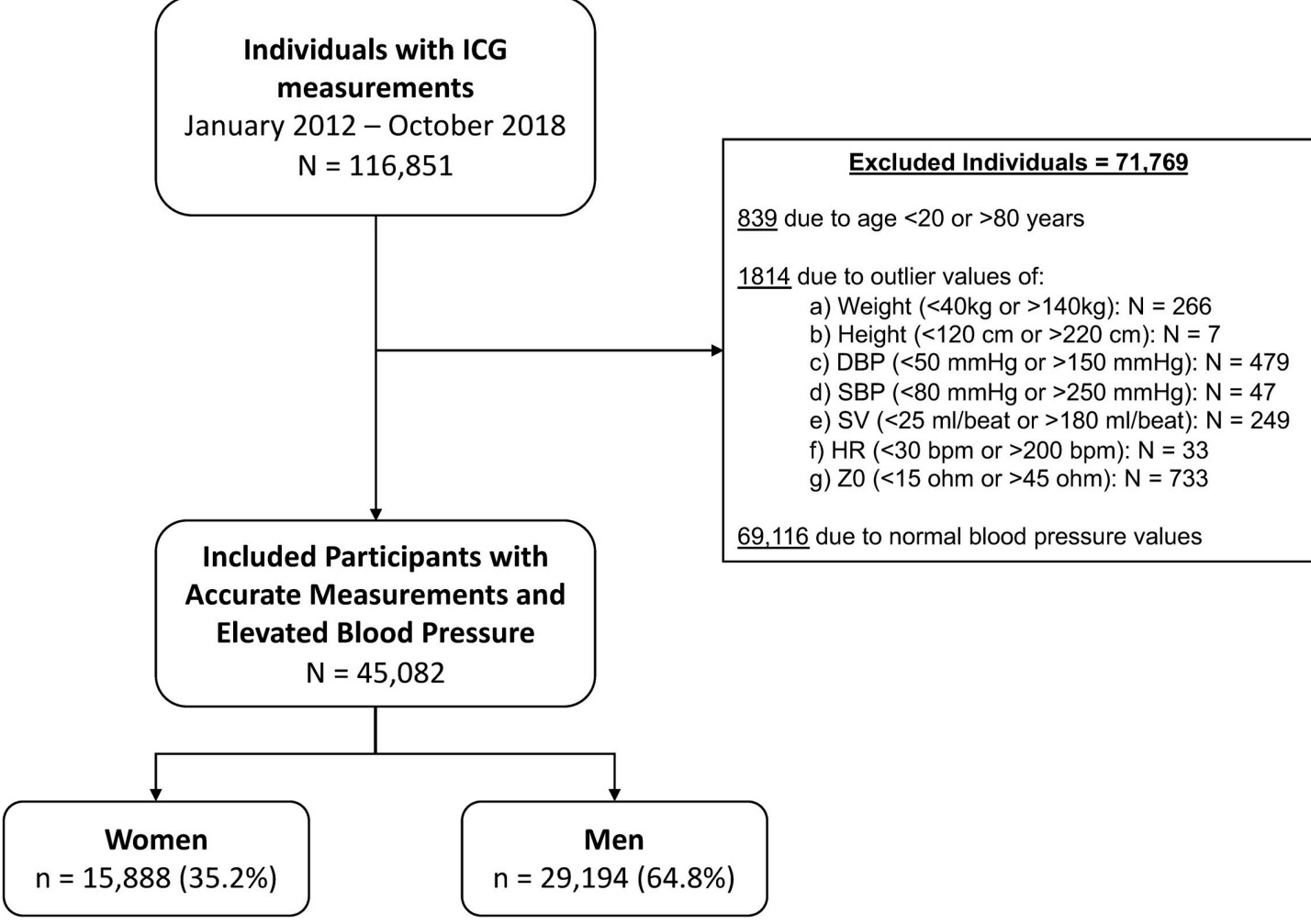

**Fig 1. Study population flowchart.** Abbreviations: ICG, impedance cardiography; DBP, diastolic blood pressure; SBP, systolic blood pressure; SV, stroke volume; HR, heart rate; Z0, baseline impedance.

and obesity as $\geq$27.5 kg/m$^2$ [35]. We defined a predominantly vascular hypertension phenotype as high SVRI (>2400 dynes·sec·cm$^{-5}$·m$^2$) with a low or normal CI (<2.5 L/min/m$^2$ or 2.5–4 L/min/m$^2$, respectively), and predominantly cardiac hypertension phenotype as high CI (>4 L/min/m$^2$) with low or normal SVRI (<2000 dynes·sec·cm$^{-5}$·m$^2$ or 2000–2400 dynes·sec·cm$^{-5}$·m$^2$, respectively) [11,36,37].

## Statistical analysis

We calculated means with standard deviations (SD) for continuous variables and frequencies for categorical variables, and assessed for the significance of the inter-group differences using ANOVA and Chi-square test (with Yates' correction), respectively. Next, the relationship between these parameters and age, BMI and SBP was evaluated using least squares method (LMS) curves [38]. To assess the association of sex with CO, CI, SVR, and SVRI we used unadjusted and sequential adjusted linear regression models and reported the female sex beta coefficient and its respective 95% confidence interval (95% CI) for the entire study population, among those <50 years old, and among those $\geq$50 years old. The sequential adjusted models

were built as follows: adjusted model 1 included age and region; model 2 included variables from model 1 plus BMI. Finally, we used density plots to characterize the distribution of the hemodynamic parameters by sex across the different strata, estimating the percentage of the plot area that overlaps between women and men on each stratum. To account for potential residual confounding, we performed a nearest neighbor propensity score matching sensitivity analysis, using region, age, SBP, DBP, and BMI. Propensity score generation and 1:1 match for samples between men and women groups were performed using the MatchIt package in R [39]. For reproducibility and comparison with prior studies, and aligned with Chinese hypertension guidelines cutoff blood pressure values [40], we also performed a sensitivity analysis replicating these analyses on a subpopulation of individuals with SBP ≥140 mmHg or DBP ≥90 mmHg.

All statistical analyses were conducted using R, version 3.6.2 (The R Foundation for Statistical Computing). Statistical significance was defined as a 2-tailed P<0.05. Coauthors JG, HZ, and ZJM take responsibility for the analysis.

### Ethics statement

This project received an exemption from review from the Institutional Review Board at Yale School of Medicine and at Shanghai Jiao Tong University College of Biotechnology as we used de-identified data provided by the iKang Health group. Given that the de-identified data were provided by a third party, we did not need to collect consent for participation.

## Results

### Age, body mass index, and hemodynamic variables and phenotypes by sex

We included 45,082 individuals with elevated blood pressure, of which 15,888 (35.2%) were women. Overall, women had a higher mean age than men (54.5 [±11.8] years vs 48.0 [±13.0] years, P<0.001) and were less likely to be obese (17.2% vs 23.5%, P<0.001) (Table 1). Women had a higher mean SBP than men (139.0 [±15.7] mmHg vs 136.8 [±13.8] mmHg, respectively, P<0.001), but a lower mean DBP (82.6 [±9.0] mmHg vs 85.6 [±8.9] mmHg, P< 0.001). Among those <50 years of age, women had lower mean SBP and DBP compared with men of the same age group (P<0.001 for each), whereas among those older than 50 years they had higher mean SBP and lower mean DBP than men (P<0.001 for each) (**Table 1**).

Overall, women had lower mean CO and CI and higher mean SVR and SVRI than men (P<0.001 for all). When stratified by age, women <50 years old had a higher mean CI and a lower mean SVRI than men (P< 0.001 for both), whereas, among those older than 50 years, women had a lower mean CI (P< 0.001) and a higher mean SVRI (P<0.001) (**Table 1**). Also, compared with men, women were more likely to have predominantly vascular hemodynamic hypertension phenotype (61.6% vs 56.2%, P<0.001) but only slightly less likely to have a predominantly cardiac phenotype (16.11% vs 17.8%, P<0.001). Compared with men of the same age, women <50 years old were less likely to have a predominantly vascular phenotype and more likely to have a predominantly cardiac phenotype (38% vs 46.4% and 31.5% vs 22.5%, respectively, P<0.001 for each comparison). Women older than 50 years, on the other hand, compared with men of the same age group, were more likely to have a predominantly vascular phenotype (70.5% vs 67.2%, P<0.001) and less likely to have a predominantly cardiac phenotype (10.2% vs 12.4%, P<0.001) (**Table 1**). Similar results were found in the sensitivity analysis among those with SBP ≥140 mmHg or DBP ≥90 mmHg, although in this subpopulation there was no significant difference in the mean CI and SVRI between men and women <50 years (**S1 Table**).

**Table 1. Sex differences in clinical and hemodynamic variables by age group among adults with elevated blood pressure.**

| | All | | | <50 years old | | | ≥50 years old | | |
|---|---|---|---|---|---|---|---|---|---|
| | Women N = 15,888 | Men N = 29194 | P value | Women N = 4,384 | Men N = 15,512 | P value | Women N = 11,504 | Men N = 13,682 | P value |
| Age (years) | 54.5 (11.8) | 48.0 (13.0) | <0.001 | 39.3 (8.1) | 38.0 (7.3) | <0.001 | 60.2 (6.9) | 59.5 (7.2) | <0.001 |
| BMI (kg/m2) | 24.4 (3.5) | 25.5 (3.2) | <0.001 | 23.5 (3.7) | 25.7 (3.4) | <0.001 | 24.8 (3.3) | 25.2 (3.0) | <0.001 |
| Obesity* | 2733 (17.2%) | 6858 (23.49%) | <0.001 | 585 (13.34%) | 4057 (26.15%) | <0.001 | 2148 (18.67%) | 2801 (20.47%) | <0.001 |
| Region | | | <0.001 | | | <0.001 | | | <0.001 |
| East | 5965 (37.54%) | 13275 (45.47%) | | 1956 (20.86%) | 7419 (79.14%) | | 4009 (40.64%) | 5856 (59.36%) | |
| North | 3665 (23.07%) | 4156 (14.24%) | | 779 (29.59%) | 1854 (70.41%) | | 2886 (55.63%) | 2302 (44.37%) | |
| South | 2737 (17.23%) | 4317 (14.79%) | | 686 (22.66%) | 2341 (77.34%) | | 2051 (50.93%) | 1976 (49.07%) | |
| Southwest | 3521 (22.16%) | 7446 (25.51%) | | 963 (19.81%) | 3898 (80.19%) | | 2558 (41.89%) | 3548 (58.11%) | |
| Blood pressure (mmHg) | | | | | | | | | |
| Systolic | 139.0 (15.7) | 136.7 (13.8) | <0.001 | 131.2 (13.2) | 133.7 (12.2) | <0.001 | 142.0 (15.6) | 140.1 (14.8) | <0.001 |
| Diastolic | 82.6 (9.00) | 85.6 (8.9) | <0.001 | 83.3 (7.9) | 85.3 (8.9) | <0.001 | 82.4 (9.4) | 86.01 (9.0) | <0.001 |
| Hypertension phenotype | | | | | | | | | |
| Predominantly cardiac[†] | 2559 (16.11%) | 5185 (17.76%) | <0.001 | 1382 (31.52%) | 3495 (22.53%) | <0.001 | 1177 (10.23%) | 1690 (12.35%) | <0.001 |
| Predominantly vascular[‡] | 9780 (61.56%) | 16396 (56.16%) | <0.001 | 1666 (38.00%) | 7197 (46.40%) | <0.001 | 8114 (70.53%) | 9199 (67.23%) | <0.001 |
| Low/normal CI & low/normal SVRI | 3531 (22.22)% | 7548 (25.86%) | <0.001 | 1,330 (30.34%) | 4,790 (30.88%) | 0.50 | 2,201 (19.13%) | 2,758 (20.16%) | 0.04 |
| High CI & high SVRI | 18 (0.11%) | 65 (0.22%) | 0.01 | 6 (0.14%) | 30 (0.19%) | 0.56 | 12 (0.10%) | 35 (0.26%) | 0.01 |
| ICG parameters | | | | | | | | | |
| Heart rate (bpm) | 69.4 (11.4) | 69.5 (11.3) | 0.63 | 72.6 (11.9) | 70.8 (11.1) | <0.001 | 68.2 (10.9) | 68.0 (11.3) | 0.15 |
| Stroke volume (mL) | 72.9 (18.6) | 88.8 (21.5) | <0.001 | 80.0 (18.8) | 93.0 (21.6) | <0.001 | 70.2 (17.7) | 84.0 (20.5) | <0.001 |
| CO (L/min) | 5.0 (1.4) | 6.1 (1.5) | <0.001 | 5.8 (1.4) | 6.5 (1.4) | <0.001 | 4.7 (1.2) | 5.6 (1.4) | <0.001 |
| CI (L/min/m$^2$) | 3.2 (0.8) | 3.3 (0.8) | <0.001 | 3.6 (0.9) | 3.5 (0.7) | <0.001 | 3.0 (0.8) | 3.2 (0.7) | <0.001 |
| SVR (dynes·sec·cm$^{-5}$) | 1744 (523) | 1433 (389) | <0.001 | 1471 (411) | 1315 (324) | <0.001 | 1848 (524) | 1565 (412) | <0.001 |
| SVRI (dynes·sec·cm$^{-5}$·m$^2$) | 2734.1 (809.9) | 2596.3 (677.2) | <0.001 | 2326.0 (658.0) | 2435.1 (598.6) | <0.001 | 2889.6 (808.3) | 2779.2 (713.8) | <0.001 |

Data are presented as mean (SD) for continuous variables and n (%) for categorical variables.

* Obesity was defined as BMI ≥27.5 kg/m2.

† A predominantly cardiac hypertension phenotype was determined by high CI with low or normal SVRI.

‡ Predominantly vascular hypertension phenotype was determined by low or normal CI with high SVRI.

Abbreviations: SD = Standard Deviation, BMI = Body Mass Index, ICG = Impedance Cardiography, SVR = Systemic Vascular Resistance, SVRI = Systemic Vascular Resistance Index, CO = Cardiac Output, CI = Cardiac Index.

### Relationship of hemodynamic variables and age among women and men

Plots of median CO, CI, SVR, and SVRI with age are presented in **Fig 2**. With age, median CO decreased for both sexes, being consistently lower among women (**Fig 2A**). Although median CI also decreased with age for both sexes, it was higher in women before age 50 than men of the same age, becoming similar afterwards (**Fig 2B**). On the other hand, SVR increased with age in both sexes, being consistently higher among women (**Fig 2C**). Although SVRI increased in both groups, it was lower among women compared with men among individuals <50 years

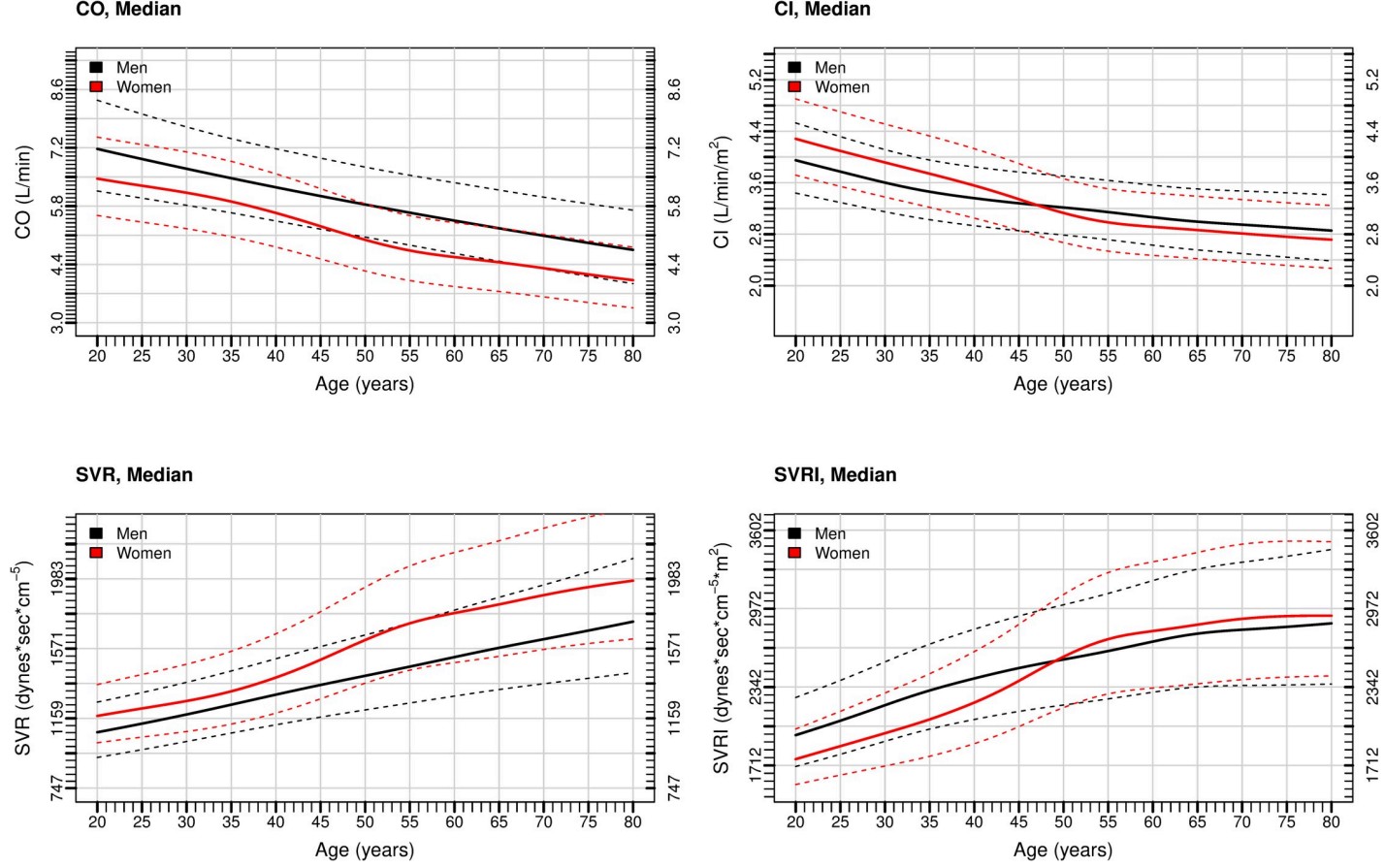

**Fig 2.** Median Cardiac Output (A), Cardiac Index (B), Systemic Vascular Resistance (C), and Systemic Vascular Resistance Index (D) by Age Among Women and Men with Elevated Blood Pressure. Solid lines represent the median. Dashed lines represent the 25th and 75th percentile. Abbreviations: CO, cardiac output; CI, cardiac index; SVR, systemic vascular resistance; SVRI, systemic vascular resistance index.

old, having a steeper increase with age among young women and becoming similar between the 2 groups among individuals ≥50 years of age (**Fig 2D**). These results were consistent with those observed when analyzing individuals with SBP ≥140 mmHg or ≥90 mmHg only (**S1 Fig**).

## Multivariable linear regression and propensity score matching sensitivity analysis

Unadjusted and sequentially-adjusted female sex beta coefficients (β) for CO, CI, SVR, and SVRI are presented in **Table 2**. After adjusting for age, BMI, and region, female sex was associated with lower cardiac output overall (β = -0.78 [95% CI: -0.8, -0.75]), among those <50 years old (β = -0.59 [95% CI: -0.64, -0.54]), and among those ≥50 years old (β = -0.86 [95% CI: -0.89, -0.83]). However, female sex was positively associated with CI only among those <50 years (β = 0.07 [95% CI: 0.04, 0.09]), having a negative association overall (β = -0.08 [95% CI: -0.09, -0.06]) and among those ≥50 years of age (β = -0.15 [95% CI: -0.16, -0.13]). On the other hand, female sex was associated with higher SVR overall (β = 229 [95% CI: 221, 238]), among those <50 years (β = 140 [95% CI: 128, 151]), and among those ≥50 years of age (β = 274 [95% CI: 262, 285]). Lastly, female sex had a negative association with SVRI only among those <50 years old (β = -31.7 [95% CI: -51.2, -12.2]), having a positive association among

**Table 2.  Unadjusted and sequentially-adjusted association of female sex with cardiac output, cardiac index, systemic vascular resistance, and systemic vascular resistance index, overall and by age categories.**

| Hemodynamic Variable | Female Sex β Coefficient (95% CI) | | |
|---|---|---|---|
| | Unadjusted Model | Adjusted Model 1[*] | Adjusted Model 2[†] |
| **Cardiac Output,** (L/min) | | | |
| Overall | -1.07 (-1.1, -1.04) | -0.79 (-0.82, -0.77) | -0.78 (-0.8, -0.75) |
| <50 years old | -0.73 (-0.78, -0.68) | -0.67 (-0.71, -0.62) | -0.59 (-0.64, -0.54) |
| ≥50 years old | -0.90 (-0.93, -0.87) | -0.86 (-0.89, -0.83) | -0.86 (-0.89, -0.83) |
| **Cardiac Index,** (L/min/m$^2$) | | | |
| Overall | -0.15 (-0.16, -0.13) | -0.01 (-0.03, 0)[‡] | -0.08 (-0.09, -0.06) |
| <50 years old | 0.14 (0.12, 0.17) | 0.19 (0.16, 0.21) | 0.07 (0.04, 0.09) |
| ≥50 years old | -0.14 (-0.16, -0.12) | -0.12 (-0.14, -0.1) | -0.15 (-0.16, -0.13) |
| **Systemic Vascular Resistance,**(dynes·sec·cm$^{-5}$) | | | |
| Overall | 312 (303, 320) | 225 (216, 233) | 230 (221, 238) |
| <50 years old | 156 (144, 167) | 138 (127, 149) | 140 (128, 151) |
| ≥50 years old | 282 (271, 294) | 271 (260, 282) | 274 (262, 285) |
| **Systemic Vascular Resistance Index,** (dynes·sec·cm$^{-5}$·m$^2$) | | | |
| Overall | 137.8 (123.7, 151.8) | 6.0 (-7.6, 19.7)[§] | 73.5 (60.2, 86.8) |
| <50 years old | -109.1 (-129.7, -88.6) | -146.9 (-166.6, -127.3) | -31.7 (-51.2, -12.2) |
| ≥50 years old | 110.5 (91.7, 129.3) | 88.5 (69.9, 107.2) | 120.4 (102.4, 138.5) |

[*] Model 1 was adjusted for age and region.

[†] Model 2 was adjusted for age, region, and body mass index.

[‡]P value = 0.16.

[§]P value = 0.39.

All other P values <0.001.

those >50 years of age (β = 120.4 [95% CI: 102.4, 138.5]) and among the entire study population (β = 73.5 [95% CI: 60.3, 86.8]). The direction and magnitude of these findings were mostly consistent with the ones from the adjusted model that did not include BMI (**Table 2**).

These hemodynamic differences remained in the propensity score matching sensitivity analysis (**S2 Table**) and in the sensitivity analysis among those with SBP ≥140 mmHg or DBP ≥90 mmHg (**S3 Table**), although the magnitude of the SVRI difference among those younger than 50 years was not significant.

## Hemodynamic variables distribution overlap between women and men

Density plots of CO and CI, by sex and age are shown in **Fig 3**. Overall, the CO showed a distribution overlap of 52.1% between women's and men's density plots (**Fig 3A**). Among those younger than 50 years, women's CO distribution was slightly shifted to the left compared to men's, with a 65.4% overlap between sexes (**Fig 3B**). On the other hand, among those older than 50 years, CO distribution among women had higher kurtosis and was shifted to the left when compared with men, with an overlap between sexes of 55.6% (**Fig 3C**). The indexed variable (CI) distribution showed a greater overlap between sexes, reaching 79.6% overall, 80.8% among those <50 years old, and 82.6% among those ≥50 years old (**Fig 3D–3F**, respectively).

Density plots of SVR and SVRI by sex and age are shown in **Fig 4**. Distribution of SVR showed an overlap between men and women of 56.7%, with women's density plot having lower kurtosis and shifted to the right compared with men's (**Fig 4A**). Among those <50 years old, women's SVR distribution had less kurtosis and was slightly shifted to the right compared with men's, with an overlap between sexes of 72.1% (**Fig 4B**). On the other hand, among those

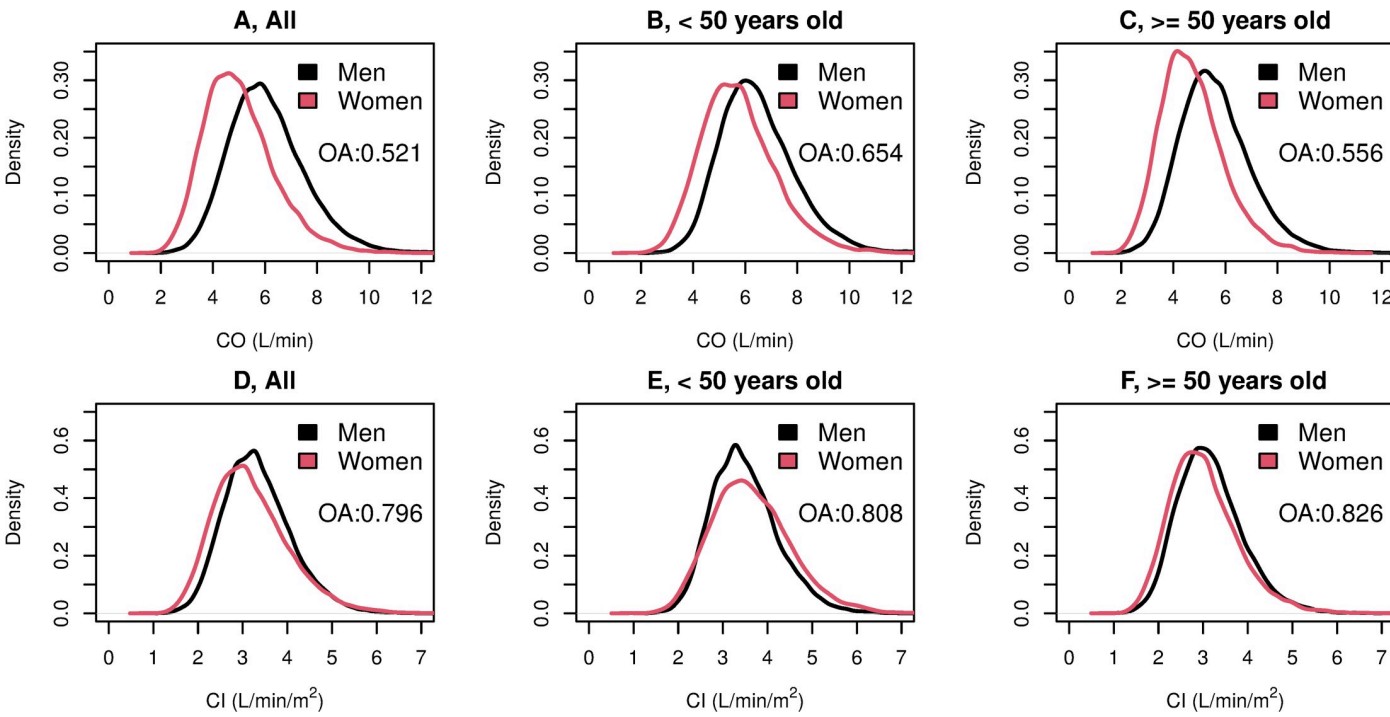

**Fig 3.** Cardiac Output (A, B, and C) and Cardiac Index (D, E, and F) Density Plots Overlap Between Women And Men with Elevated Blood Pressure by Age Category. Abbreviations: CO, cardiac output; CI, cardiac index; OA, overlapping area.

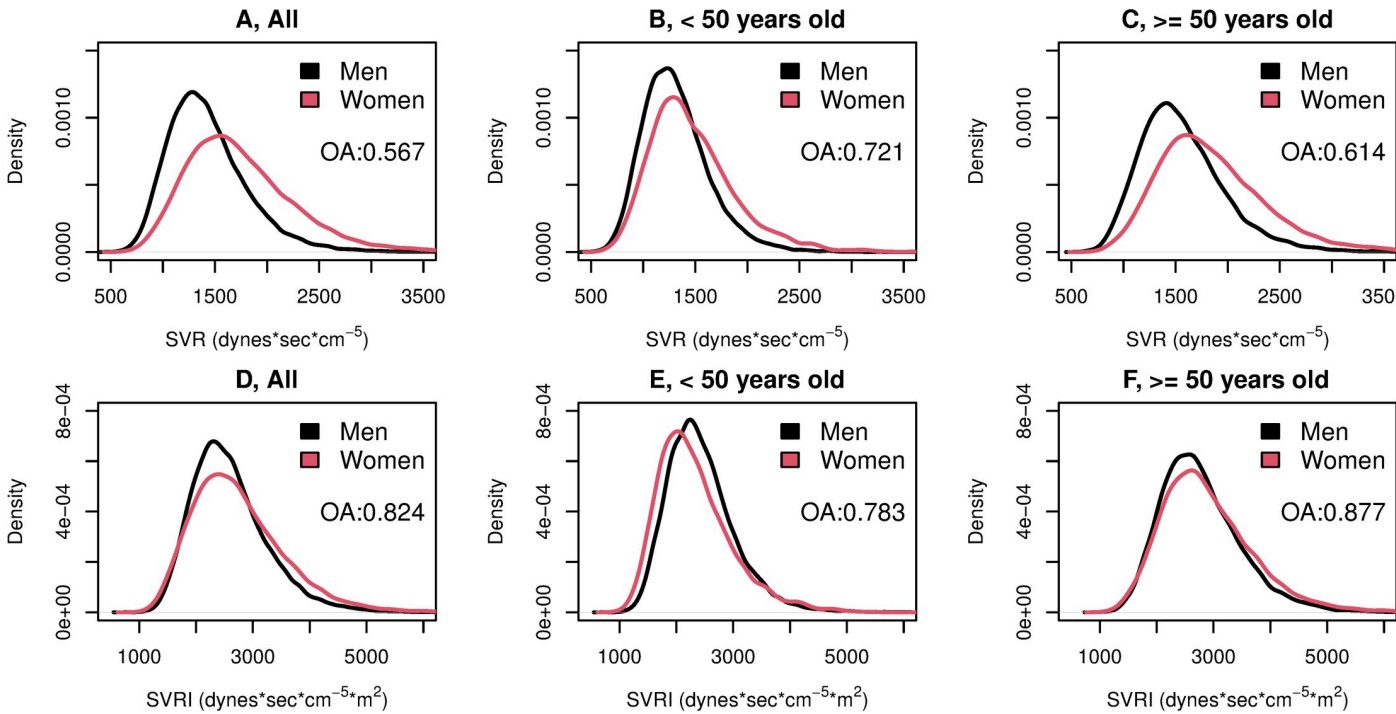

**Fig 4.** Systemic Vascular Resistance (A, B, and C), and Systemic Vascular Resistance Index (D, E, and F) Density Plots Overlap Between Women And Men with Elevated Blood Pressure by Age Category. Abbreviations: SVR, systemic vascular resistance; SVRI, systemic vascular resistance index; OA, overlapping area.

≥50 years old, women's SVR distribution was shifted to the right compared with men's, with an overlap of 61.4% between the 2 groups (**Fig 4C**). The indexed variable (SVRI), also increased in overlap between both sexes, reaching 82.4%, 78.3%, and 87.7% among the entire study population, among those <50 years old, and among those ≥50 years old, respectively (**Fig 4D–4F**, respectively).

When analyzing only those with SBP ≥140 mmHg or DBP ≥90 mmHg, the overlapping areas of the distribution of CO, CI, SVR, and SVRI were highly consistent with the ones from our main analysis (**S2 and S3 Figs**).

## Discussion

In our study, we investigated sex differences in hemodynamic variables, mainly CI and SVRI, among adults presenting with elevated blood pressure. We found that, on average, young women have higher mean CI and lower SVRI than age-matched men, although the magnitude of the SVRI difference was significantly reduced when accounting for confounders. Notably, we also found that these hemodynamic differences between sexes were reversed among those older than 50 years of age, the mean age of menopause in China [34], with women having lower mean CI and higher mean SVRI than men. Nonetheless, a key finding of our study was that, despite these overall differences, there is substantial hemodynamic heterogeneity within individuals of the same sex and age: the overlapping area of the distribution plots of CI and SVRI ranged from nearly 80% among those younger than 50 years to nearly 90% among those older than 50 years.

Our results expand the existing knowledge in 2 major ways. First, while most studies have been performed on small samples or among young individuals [5–7], our study allowed us to describe the hemodynamic sex differences in a much larger sample of individuals with elevated blood pressure, including older individuals. To the best of our knowledge this is the largest study that has compared hemodynamic variables and phenotypes between women and men across different age groups. Doing so is instrumental in these studies because of the cardiovascular risk changes associated with menopause [18,19,34,41]. Second, our study is the first to estimate the full distribution of hemodynamic parameters by sex, rather than being limited to comparing the average values. This approach allowed us to describe for the first time that–besides significant average differences–sex is not a reliable indicator of the individual hemodynamic phenotype, particularly among those older than 50 years old where the distributions of CI and SVRI among women and men were almost identical.

Despite the geographical, sample size, and inclusion criteria differences, our results complement and are consistent with the findings from other studies in non-Asian populations [5–7] in which the authors found that, among hypertensive individuals, men had a higher CO and lower SVR than women. However, considering the well-known body composition differences between men and women [42], we analyzed the hypertension phenotype using the body surface area adjusted values (CI and SVRI) and stratified by age, increasing the comparability of these variables between sexes, consistent with our previous studies [43,44]. We found that despite the differences in mean CO and SVR, among those <50 years old hypertensive women were more likely to have a predominantly cardiac phenotype (high CI with normal/low SVRI) and less likely to have a predominantly vascular phenotype (high SVRI with normal/low CI) compared with men of the same age. Interestingly, among those older than 50 years, women were more likely to present a predominantly vascular phenotype than men. One plausible explanation for such an observation is that, along with hormonal and environmental factors [19], young women have a blunted alpha-adrenergic vasoconstriction response because of an increased beta-adrenergic vasodilatation [45] that disappears after menopause and that is

absent in men [46]. This, however, does not fully explain the association between sex and blood pressure: compared with men, women have a steeper increase in SBP thorough life, even decades before menopause [47]. Altogether, such findings might suggest that the underlying mechanisms of elevated blood pressure might differ by sex, particularly among young individuals. The therapeutically implications of these sex differences are limited because of the high same-sex heterogeneity in these parameters that we observed. Beyond its potential implications for treatment adjustment or initiation [8–10], understanding if these differences in hypertension phenotypes are implicated in the known sex differences in terms of risk of subsequent cardiovascular outcomes [48] is still uncertain and deserves further investigation. Furthermore, there is also a need for longitudinal studies that help us to understand how chronic exposure to different hypertension phenotypes is associated with clinical outcomes, and if there are sex-differences in such associations.

Our study also has important implications for personalizing the care of patients presenting to an outpatient clinic with elevated blood pressure. As hypertension remains as one of the biggest public health challenges worldwide [49], there is urgency in determining possible ways of improving its treatment efficacy by using personalized therapies tailored to each patient's characteristics. Although we and other studies have shown that, on average, there are significant hemodynamic differences by sex, we also found that there is substantial same-sex heterogeneity in the hemodynamic profile. Notably, as hypertension prevalence and arterial stiffening increases among women after menopause [15,18,50], the distribution of CI and SVRI among those older than 50 years was almost the same between women and men. Thus, our study indicates that sex alone is not a good proxy of the underlying hemodynamic patterns of patients with elevated blood pressure and should not be used clinically to assume the hypertension phenotype. Instead, it is necessary to measure these parameters directly if information about the hemodynamic profile is considered necessary to guide the antihypertensive therapy.

## Limitations

The results from our study should be interpreted in light of the following limitations. First, our findings only represent a snapshot of an individual's hemodynamic status, preventing us from assessing longitudinal clinical outcomes and pathologic adaptations that may occur with long-term exposure to a particular hypertension phenotype, including structural and physiological changes in heart and vessels [51,52].

Second, although we had highly detailed hemodynamic information of each individual, the sociodemographic and clinical information (including comorbidities that may affect hemodynamic status) available to our analyses was limited. Of great importance is the lack of information regarding current antihypertensive medications usage, which could alter the hemodynamic phenotype and inclusion criteria (e.g. beta-blockers lowering CO or a patient with controlled hypertension not being included). However, studies have shown that hypertension treatment and blood pressure control rates are low in China [53–55], though they are higher among women than in men, suggesting that the majority of the patients in our study would be untreated and that most individuals with hypertension would have been included. Nonetheless, our results represent participants' hemodynamic status at the time of examination in an outpatient clinic, regardless of their medications, which could help guide the treatment initiation or adjustment based on the underlying hemodynamic derangement among patients with uncontrolled blood pressure. Additionally, although menopause is a key determinant of blood pressure regulation, we did not have this self-reported event from our participants. However, for our analysis we used the average age of menopause in China and the results were consistent with this hemodynamic shift among women around age 50 years.

Lastly, we did not have information on participants' smoking status or smoking history, factors associated with hypertension and increased arterial stiffness, although the latter is less certain [56,57]. Smoking is one of the major public health challenges among Chinese men, with a prevalence of nearly 1 in 2 among them and only around 2% among women [58,59]. The impact of smoking on the hypertension phenotype and if such an association is modified by sex remains to be studied in detail.

Third, a single BP measurement was recorded per participant, which could have affected the precision in this variable. However, all study centers followed the same standardized protocol for men and women, as described in the Methods section, to reduce inter-observer variability. Moreover, such a compromise in precision would most likely shift the observed sex differences towards the null rather than towards significance.

Fourth, although we performed a robust main analysis, which included adjustment for multiple confounders, and a sensitivity analysis that used propensity score matching, we cannot rule out that the differences observed between men and women were due to residual confounding.

## Conclusions

Women and men with elevated blood pressure in China have differences in the average values of the hemodynamic determinants of blood pressure. However, the magnitude of such differences is significantly reduced with age and there is substantial overlap in the distribution of the hemodynamic variables. This indicates that sex alone is insufficient to infer the underlying hypertension phenotype.

## Supporting information

**S1 Fig. Median cardiac output, cardiac index, systemic vascular resistance, and systemic vascular resistance index by age among women and men with systolic blood pressure ≥140 mmHg or diastolic blood pressure ≥90 mmHg.**
(PDF)

**S2 Fig. Cardiac output and cardiac index density plots overlap between women and men with systolic blood pressure ≥140 mmHg or diastolic blood pressure ≥90 mmHg, by age category.**
(PDF)

**S3 Fig. Systemic vascular resistance and systemic vascular resistance index density plots overlap between women and men with systolic blood pressure ≥140 mmHg or diastolic blood pressure ≥90 mmHg, by age category.**
(PDF)

**S1 Table. Sex differences in clinical and hemodynamic variables by age group among adults with systolic blood pressure ≥140 mmHg or diastolic blood pressure ≥90 mmHg.**
(PDF)

**S2 Table. Sex differences in clinical and hemodynamic variables by nearest neighbor propensity score matched subgroups.**
(PDF)

**S3 Table. Unadjusted and sequentially-adjusted association of female sex with cardiac output, cardiac index, systemic vascular resistance, and systemic vascular resistance index, overall and by age categories among adults with systolic blood pressure ≥140 mmHg or**

**diastolic blood pressure ≥90 mmHg, by age category.**
(PDF)

## Acknowledgments

The authors would like to acknowledge the support of Mr. Zhang Ligang, CEO of iKang Health Group, for the permission to use the data.

## Author Contributions

**Conceptualization:** César Caraballo, Shiwani Mahajan, Yuan Lu, Erica S. Spatz, Rachel P. Dreyer, MaoZhen Zhang, NingLing Sun, Yihong Ren, Xin Zheng, Hongyu Zhao, Hui Lu, Zheng J. Ma, Harlan M. Krumholz.

**Data curation:** Jianlei Gu, Zheng J. Ma.

**Formal analysis:** Jianlei Gu, Hongyu Zhao, Zheng J. Ma.

**Investigation:** Shiwani Mahajan, Zheng J. Ma.

**Methodology:** César Caraballo, Jianlei Gu, Hongyu Zhao, Harlan M. Krumholz.

**Project administration:** César Caraballo, Shiwani Mahajan, Harlan M. Krumholz.

**Resources:** Zheng J. Ma.

**Supervision:** Erica S. Spatz, Harlan M. Krumholz.

**Validation:** Hongyu Zhao, Zheng J. Ma.

**Visualization:** Jianlei Gu.

**Writing – original draft:** César Caraballo.

**Writing – review & editing:** Shiwani Mahajan, Jianlei Gu, Yuan Lu, Erica S. Spatz, Rachel P. Dreyer, MaoZhen Zhang, NingLing Sun, Yihong Ren, Xin Zheng, Hongyu Zhao, Hui Lu, Zheng J. Ma, Harlan M. Krumholz.

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
