## [Decision Letter · Decision Letter 0]

6 Jan 2022

PONE-D-21-16564

Hemodynamic Differences Between Women and Men with Elevated Blood Pressure in China

PLOS ONE

Dear Dr. Krumholz,

Thank you for submitting your manuscript to PLOS ONE. After careful consideration, we feel that it has merit but does not fully meet PLOS ONE’s publication criteria as it currently stands. Therefore, we invite you to submit a revised version of the manuscript that addresses the points raised during the review process.

We look forward to receiving your revised manuscript.

Kind regards,

Johannes Vogel

Academic Editor

PLOS ONE

Journal Requirements:

2. Thank you for stating the following financial disclosure: "This study was self-funded."

"Yuan Lu is supported by the National Heart, Lung, and Blood Institute (K12HL138037) and the Yale Center for Implementation Science. She was a recipient of a research agreement, through Yale University, from the Shenzhen Center for Health Information for work to advance intelligent disease prevention and health promotion. Erica S. Spatz receives support from the Food and Drug Administration to support projects within the Yale-Mayo Clinic Center of Excellence in Regulatory Science and Innovation (CERSI); the National Institute on Minority Health and Health Disparities (U54MD010711-01) to study precision-based approaches to diagnosing and preventing hypertension; and the National Institute of Biomedical Imaging and Bioengineering (R01EB028106-01) to study a cuff-less blood pressure device. Xin Zheng is supported by the CAMS Innovation Fund for Medical Science (2016-I2M-1-006), the National Key Research and Development Program (2016YFE0103800) from the Ministry of Science and Technology of China. Hui Lu is supported by the National Key R&D Program of China (2018YFC0910500) and Neil Shen’s SJTU Medical Research Fund. Zheng J. Ma is affiliated with Beijing Li-Heng Medical Technologies, Ltd, which designed the ICG device used in this study. Harlan Krumholz works under contract with the Centers for Medicare & Medicaid Services to support quality measurement programs; was a recipient of a research grant, through Yale, from Medtronic and the U.S. Food and Drug Administration to develop methods for post-market surveillance of medical devices; was a recipient of a research grant with Medtronic and is the recipient of a research grant from Johnson & Johnson, through Yale University, to support clinical trial data sharing; was a recipient of a research agreement, through Yale University, from the Shenzhen Center for Health Information for work to advance intelligent disease prevention and health promotion; collaborates with the National Center for Cardiovascular Diseases in Beijing; receives payment from the Arnold & Porter Law Firm for work related to the Sanofi clopidogrel litigation, from the Ben C. Martin Law Firm for work related to the Cook IVC filter litigation, and from the Siegfried and Jensen Law Firm for work related to Vioxx litigation; chairs a Cardiac Scientific Advisory Board for UnitedHealth; was a participant/participant representative of the IBM Watson Health Life Sciences Board; is a member of the Advisory Board for Element Science, the Advisory Board for Facebook, and the Physician Advisory Board for Aetna; and is the founder of HugoHealth, a personal health information platform, and co-founder of Refactor Health, an enterprise healthcare AI-augmented data management company. The other co-authors report no potential competing interests."

5. Please note that in order to use the direct billing option the corresponding author must be affiliated with the chosen institute. Please either amend your manuscript to change the affiliation or corresponding author, or email us at plosone@plos.org with a request to remove this option.

6. Thank you for stating the following in the Financial Disclosure section: "This study was self-funded."

We note that one or more of the authors are employed by a commercial company: "iKang Healthcare Group, Inc., Shanghai, China and Beijing Li-Heng Medical Technologies, Ltd, Beijing, China.

Reviewers' comments:

Reviewer's Responses to Questions

**Comments to the Author**

1. Is the manuscript technically sound, and do the data support the conclusions?

Reviewer #1: Partly

Reviewer #2: Partly

2. Has the statistical analysis been performed appropriately and rigorously? 

Reviewer #1: Yes

Reviewer #2: N/A

3. Have the authors made all data underlying the findings in their manuscript fully available?

Reviewer #1: No

Reviewer #2: No

4. Is the manuscript presented in an intelligible fashion and written in standard English?

Reviewer #1: Yes

Reviewer #2: No

5. Review Comments to the Author

Reviewer #1: We have made a review for manuscript number: PONE-D-21-16564, entitled;

Hemodynamic Differences Between Women and Men with Elevated Blood Pressure in China. We have some important comments:

1- It should be clear in the title and abstract that the topic is about non invasive hemodynamic data assessment.

2- A flow chart is needed.

3- Thd authors should give strong reason of why there is a restriction on the availability of data.

4- In the methodology, the diagnosis of elevated blood pressure relies only on single blood pressure measurement. This is the major concern of the current analysis. Single measurement by automated device is not enough to classify patients regarding their blood pressure level. This point should be properly discussed and defended.

5- The authors should justify the use of ICG, the authors should mention the guidelines recommendations or FDA or international medical institutes that approve the use of ICG for such purposes.

6- In the results, there was significant difference in the age of women versus men (i.e., 54 vs 45, p <0.001). This significant difference still presents after subgroups analysis of <50 yr or > 50 yr (i.e., p<0.001). This important point should be defended and discuss as significant differences in age would be important confounder and obtaining reliable results needs proper adjustment of the analysis in regard to age.

7- The current manuscript does not have data regarding long term follow up to see if the current hemodynamic differences have clinical impact on long term prognosis or not. This point should be properly defended.

8- The clinical implications of the current study should be clear and practical.

Regards

Reviewer #2: -The cofounders missing some important factors as presence of diabetes , the types of hypertension ,the risk factors as dyslipidemia and the existence of chronic kidney disease .all these factors can contribute in risk stratification and further management of hypertension.

-the age 50 is not clear as not standard for old age as WHO recommendations.

- in table 1 what is meant by predominantly cardiac

6. PLOS authors have the option to publish the peer review history of their article (what does this mean?). If published, this will include your full peer review and any attached files.

Reviewer #1: **Yes: **Rami Riziq Yousef Abumuaileq

Reviewer #2: No

---

## [Author Response · Author response to Decision Letter 0]

12 Apr 2022

Response To Reviewers

PONE-D-21-16564: Hemodynamic Differences Between Women and Men with Elevated Blood Pressure in China

We appreciate the academic editor’s and reviewers’ comments, which have helped us to improve our manuscript. We have responded to each comment (reproduced in bold) and detailed our changes to the manuscript. In this document, quoted text is presented indented and new additions (or relevant text when specified) underlined. The page and line numbers that we reference below are based on the clean version of our revised manuscript. New references are listed at the end of each response. We first addressed the academic editor’s comments and then each of the two reviewers.

ACADEMIC EDITOR

Journal Requirements:

Response: We have reviewed the style requirements and made the necessary adjustments.

2. Thank you for stating the following financial disclosure: "This study was self-funded."

Response: Thank you. We modified our funding statement, as shown below (underlined text is new):

Source of funding: 

The authors received no specific funding for this work. 

"Yuan Lu is supported by the National Heart, Lung, and Blood Institute (K12HL138037) and the Yale Center for Implementation Science. She was a recipient of a research agreement, through Yale University, from the Shenzhen Center for Health Information for work to advance intelligent disease prevention and health promotion. Erica S. Spatz receives support from the Food and Drug Administration to support projects within the Yale-Mayo Clinic Center of Excellence in Regulatory Science and Innovation (CERSI); the National Institute on Minority Health and Health Disparities (U54MD010711-01) to study precision-based approaches to diagnosing and preventing hypertension; and the National Institute of Biomedical Imaging and Bioengineering (R01EB028106-01) to study a cuff-less blood pressure device. Xin Zheng is supported by the CAMS Innovation Fund for Medical Science (2016-I2M-1-006), the National Key Research and Development Program (2016YFE0103800) from the Ministry of Science and Technology of China. Hui Lu is supported by the National Key R&D Program of China (2018YFC0910500) and Neil Shen’s SJTU Medical Research Fund. Zheng J. Ma is affiliated with Beijing Li-Heng Medical Technologies, Ltd, which designed the ICG device used in this study. Harlan Krumholz works under contract with the Centers for Medicare & Medicaid Services to support quality measurement programs; was a recipient of a research grant, through Yale, from Medtronic and the U.S. Food and Drug Administration to develop methods for post-market surveillance of medical devices; was a recipient of a research grant with Medtronic and is the recipient of a research grant from Johnson & Johnson, through Yale University, to support clinical trial data sharing; was a recipient of a research agreement, through Yale University, from the Shenzhen Center for Health Information for work to advance intelligent disease prevention and health promotion; collaborates with the National Center for Cardiovascular Diseases in Beijing; receives payment from the Arnold & Porter Law Firm for work related to the Sanofi clopidogrel litigation, from the Ben C. Martin Law Firm for work related to the Cook IVC filter litigation, and from the Siegfried and Jensen Law Firm for work related to Vioxx litigation; chairs a Cardiac Scientific Advisory Board for UnitedHealth; was a participant/participant representative of the IBM Watson Health Life Sciences Board; is a member of the Advisory Board for Element Science, the Advisory Board for Facebook, and the Physician Advisory Board for Aetna; and is the founder of HugoHealth, a personal health information platform, and co-founder of Refactor Health, an enterprise healthcare AI-augmented data management company. The other co-authors report no potential competing interests."

Response: We reviewed the guide for authors and confirm that such competing interests do not alter our adherence to PLOS ONE policies on sharing data and materials. The revised disclosure now reads as below (underlined text is new):

 DISCLOSURES

Yuan Lu is supported by the National Heart, Lung, and Blood Institute (K12HL138037) and the Yale Center for Implementation Science. Erica S. Spatz receives support from the Food and Drug Administration to support projects within the Yale-Mayo Clinic Center of Excellence in Regulatory Science and Innovation (CERSI); the National Institute on Minority Health and Health Disparities (U54MD010711-01) to study precision-based approaches to diagnosing and preventing hypertension; and the National Institute of Biomedical Imaging and Bioengineering (R01EB028106-01) to study a cuff-less blood pressure device. Xin Zheng is supported by the CAMS Innovation Fund for Medical Science (2016-I2M-1-006) and the National Key Research and Development Program (2016YFE0103800) from the Ministry of Science and Technology of China. Hui Lu is supported by the National Key R&D Program of China (2018YFC0910500) and Neil Shen’s SJTU Medical Research Fund. Zheng J. Ma is affiliated with Beijing Li-Heng Medical Technologies, Ltd, which designed the ICG device used in this study. This affiliation did not play a role in the study design, data collection and analysis, decision to publish, or preparation of the manuscript. In the past three years, Harlan Krumholz received expenses and/or personal fees from UnitedHealth, Element Science, Aetna, Reality Labs, Tesseract/4Catalyst, F-Prime, the Siegfried and Jensen Law Firm, Arnold and Porter Law Firm, and Martin/Baughman Law Firm. He is a co-founder of Refactor Health and HugoHealth, and is associated with contracts, through Yale New Haven Hospital, from the Centers for Medicare & Medicaid Services and through Yale University from Johnson & Johnson. Such competing interests do not alter our adherence to PLOS ONE policies on sharing data and materials. The other co-authors report no potential competing interests.

Response: We updated the data availability statement as shown below after discussion with, and guidance from, the PLOS One editorial office. 

iKang Health Group provided the de-identified data used in this study to SJTU-Yale Joint Center for Biostatistics in Shanghai, China. This is where all data are stored and all analyses were performed. There are recent strict legal restrictions in publicly sharing data from China outside of its borders (see Personal Information Protection Law, November 2021). Requests can be sent to iKang Health at zhaohui.wang@ikang.com for consideration to share data in a legally compliant manner.

5. Please note that in order to use the direct billing option the corresponding author must be affiliated with the chosen institute. Please either amend your manuscript to change the affiliation or corresponding author, or email us at plosone@plos.org with a request to remove this option.

Response: We confirm that the corresponding author, Harlan M. Krumholz, is affiliated with Yale University with whom PLOS ONE has the direct billing option. We edited his affiliations in the title page.

6. Thank you for stating the following in the Financial Disclosure section: "This study was self-funded."

We note that one or more of the authors are employed by a commercial company: "iKang Healthcare Group, Inc., Shanghai, China and Beijing Li-Heng Medical Technologies, Ltd, Beijing, China.

Response: We confirm that such affiliation did not have any role in the study design, data collection, data analysis, manuscript preparation, or decision to publish, nor was any author compensated for this project in any way. 

The revised and updated disclosures are detailed in Comment 3 above.

Response: We have updated our ethics statement, as shown below.

Methods section, pages 6 and 7, lines 132-136 (underlined text is new):

Ethics statement

This project received an exemption for review from the Institutional Review Board at Yale School of Medicine and at Shanghai Jiao Tong University College of Biotechnology as we used de-identified data provided by the iKang Health group. Given that the de-identified data were provided by a third party, we did not need to collect consent for analysis. 

REVIEWER 1

Comment 1: It should be clear in the title and abstract that the topic is about non invasive hemodynamic data assessment.

Response: Thank you for your suggestion. We modified the study title, as shown below (underlined text is new)

Title: Hemodynamic Differences Between Women and Men with Elevated Blood Pressure in China: A Non-Invasive Assessment of 45,082 Adults Using Impedance Cardiography

Comment 2: A flow chart is needed.

Response: Thank you for your suggestion. We moved the study population flowchart from the supporting information document to the main text as new Figure 1 (shown below):

Figure 1. Study population flowchart

Legend: Abbreviations: ICG, impedance cardiography; DBP, diastolic blood pressure; SBP, systolic blood pressure; SV, stroke volume; HR, heart rate; Z0, baseline impedance. 

Comment 3: The authors should give strong reason of why there is a restriction on the availability of data.

Response: The data used in this study was provided by iKang Health Group to be analyzed by the SJTU-Yale Joint Center for Biostatistics in Shanghai, China. Although deidentified, data are from a third party and we are restricted from sharing it publicly. Data requests, however, can be submitted to iKang Health Group for consideration, and we made the code used to analyze the data publicly available (https://www.doi.org/10.5281/zenodo.5931975). We updated our data availability statement to reflect this, as below.

Methods, page 3, lines 51-56 (underlined text is new):

Data source

iKang Health Group provided the de-identified data used in this study to SJTU-Yale Joint Center for Biostatistics in Shanghai, China. This is where all data are stored and all analyses were performed. Requests for data can be submitted to the iKang Health Group at maozhen.zhang@ikang.com. The code used to analyze the data is publicly available at https://www.doi.org/10.5281/zenodo.5931975.

Comment 4: In the methodology, the diagnosis of elevated blood pressure relies only on single blood pressure measurement. This is the major concern of the current analysis. Single measurement by automated device is not enough to classify patients regarding their blood pressure level. This point should be properly discussed and defended.

Response: We recognize that using a single measurement is not ideal because of the possible measurement-to-measurement variability in a single individual. It is possible that we identified individuals as hypertensive based on the single measurement that would not have been included if an average of 2 or more measurements were used instead. However, we also performed a sensitivity analysis using higher blood pressure cutoffs (SBP≥140 mmHg, DBP≥90 mmHg), in which likelihood of inadequately identifying hypertensive individuals is decreased. The findings of this sensitivity analysis were consistent with those of the main analysis. 

Furthermore, in all participating sites, blood pressure was measured following a standardized protocol. Although a single measurement may compromise the precision of an individual measurement, the impact of this limitation is reduced by the fact that all sites followed the same protocol with the same calibrated device, regardless of the participants’ sex. We have expanded the limitations, as shown below:

Discussion section, page 17, lines 340-344 (underlined text is new):

Third, a single BP measurement was recorded per participant, which could have affected the precision in this variable. However, all study centers followed the same standardized protocol for men and women, as described in the Methods section, to reduce inter-observer variability. Moreover, such a compromise in precision would most likely shift the observed sex differences towards the null rather than towards significance.

Comment 5: The authors should justify the use of ICG, the authors should mention the guidelines recommendations or FDA or international medical institutes that approve the use of ICG for such purposes.

Response: The non-invasive nature of ICG measurement makes it an ideal technology for monitoring hemodynamic parameters in clinics. We leveraged data in clinics where they were being routinely used. 

Although there are no guideline recommendations for routine hemodynamic measurement using impedance cardiography, many devices that use ICG have been approved by the FDA for such purposes in different scenarios. Below are some of the FDA 510(k) premarket notifications that support the use of devices using ICG for hemodynamic assessment: 

- Approval K080941: intended to use ICG among patients with cardiovascular disease who need cardiac assessment. Available from: 

o https://www.accessdata.fda.gov/scripts/cdrh/cfdocs/cfpmn/pmn.cfm?ID=K080941

- Approval K172196: intended to use ICG among patients who are expected to be intubated for less than 24 hours. Available from:

o https://www.accessdata.fda.gov/scripts/cdrh/cfdocs/cfpmn/pmn.cfm?ID=K172196

- Approval K160899: intended to use ICG among patients with fluid-management disorders such as heart failure, chronic kidney disease, or who use diuretics. Available from:

o https://www.accessdata.fda.gov/scripts/cdrh/cfdocs/cfpmn/pmn.cfm?ID=K160899

- Approval K041434: intended to use ICG among patients in hospital areas, including clinics. Available from:

o https://www.accessdata.fda.gov/scripts/cdrh/cfdocs/cfpmn/pmn.cfm?ID=K041434

- Approval K110645: intended to use ICG for general hemodynamic monitoring. Available from: 

o https://www.accessdata.fda.gov/scripts/cdrh/cfdocs/cfpmn/pmn.cfm?ID=K110645

The device used in our study has been validated against invasive thermodilution and echocardiography in different scenarios, as mentioned in the methods section (see below).

Methods, page 5, lines 90-94:

The ICG device used (CHM T3002/P3005, designed by Beijing Li-Heng Medical Technologies, Ltd, manufactured by Shandong Baolihao Medical Appliances, Ltd.) was developed based on improved hardware and advanced digital filtering algorithms,[1] and has been validated versus both invasive thermodilution and non-invasive echocardiography in different settings.[2-4]

We defer to the editors whether to cite in the main text the FDA 510(k) documents listed above.

References:

1. Ma L. Development and application of the latest model of the cardiac hemodynamics monitoring system. International J Cardiovasc Med. 2009;10:11.

2. An X-g, Zhao Y, Gao J. Clinic evaluation of noninvasive hemodynamic monitoring in patients undergoing coronary artery surgery. Chin J Cardiovasc Rev. 2008;2:010.

3. Hong H, Jin X, Pan C, Gao X, Liu M, Jiang H, Ge J. Analysis of the correlation between non-invasive hemodynamic monitor and cardiac echocardiography on the evaluation of cardiac function. Chinese Journal of Medical Instrumentation. 2009;33:328-331.

4. Chen W, Huang D, Zeng J, Deng R, Deng G, Chen W, Zou Y. Application of noninvasive cardiac hemodynamic monitor for ICU critically ill patients. Hainan Medical Journal. 2018;29:15.

Comment 6: In the results, there was significant difference in the age of women versus men (i.e., 54 vs 45, p <0.001). This significant difference still presents after subgroups analysis of <50 yr or > 50 yr (i.e., p<0.001). This important point should be defended and discuss as significant differences in age would be important confounder and obtaining reliable results needs proper adjustment of the analysis in regard to age.

Response: Indeed, there were important differences in the mean age between women and men that could influence our findings. We believe, however, that we robustly accounted for such differences in 2 major ways, listed below:

1. We performed a sequentially adjusted linear regression, shown in Table 2. Age was included in Adjusted Model 1 and Adjusted Model 2, without significantly altering the association of female sex and the hemodynamic indicators (either overall or stratified at age 50). In these multivariable analyses, we found that women <50 years old had lower systemic vascular resistance index and higher cardiac index than men of their same age group, whereas among those ≥50 years old, women had higher systemic vascular resistance index but lower cardiac index. Please find Table 2 at the end of this document.

2. We also performed a sensitivity analysis using a nearest neighbor propensity score matching, using region, age, SBP, DBP, and BMI. In this sensitivity analysis, showed in S2 Table (also at the end of this document), findings in magnitude and direction of the multivariable linear regression were similar, with the exception of no significant difference in systemic vascular resistance index among those <50 years old. 

Thus, the observed sex differences are not likely to arise just from differences in the mean age of our study sample. Nonetheless, it should be clear that although we found statistically significant differences in the mean of the hemodynamic variables studied, the main finding is the heterogeneity in their distribution such that sex was not determinative of the hemodynamic profile, as below:

Discussion section, pages 15-16, lines 299-311 (underlined relevant text):

Our study also has important implications for personalizing the care of patients presenting to an outpatient clinic with elevated blood pressure. As hypertension remains as one of the biggest public health challenges worldwide, there is urgency in determining possible ways of improving its treatment efficacy by using personalized therapies tailored to each patient’s characteristics. Although we and other studies have shown that, on average, there are significant hemodynamic differences by sex, we also found that there is substantial same-sex heterogeneity in the hemodynamic profile. Notably, as hypertension prevalence and arterial stiffening increases among women after menopause, the distribution overlap of CI and SVRI among those older than 50 years was almost the same between women and men. Thus, our study indicates that sex alone is not a good proxy of the underlying hemodynamic patterns of patients with elevated blood pressure and should not be used clinically to assume the hypertension phenotype. Instead, it is necessary to measure these parameters directly if information about the hemodynamic profile is considered necessary to guide the antihypertensive therapy.

Nonetheless, we added this as a limitation of our analysis, as shown below:

Discussion section, page 17, lines 345-348 (underlined text is new):

Fourth, although we performed a robust main analysis, which included adjustment for multiple confounders, and a sensitivity analysis that used propensity score matching, we cannot rule out that the differences observed between men and women were due to residual confounding.

Comment 7: The current manuscript does not have data regarding long term follow up to see if the current hemodynamic differences have clinical impact on long term prognosis or not. This point should be properly defended.

Response: Although we lack such information, we agree on the importance of investigating the long-term association of different hemodynamic phenotypes. However, such an objective is outside the scope of our study which, cross-sectional in design, aimed to evaluate the differences between men and women in hemodynamic profiles in an outpatient setting. 

Discussion section, page 15, lines 289-298 (underlined text is new or edited):

Altogether, such findings might suggest that the underlying mechanisms of elevated blood pressure might differ by sex, particularly among young individuals. The clinical implications of these sex differences are limited because of the high same-sex heterogeneity in these parameters that we observed. Beyond its potential implications for treatment adjustment or initiation, understanding if these differences in hypertension phenotypes are implicated in the known sex differences in terms of risk of subsequent cardiovascular outcomes is still uncertain and deserves further investigation. Furthermore, there is a need for longitudinal studies that help understand how chronic exposure to different hypertension phenotypes is associated with clinical outcomes, and if there are sex differences in such associations.

We also edited our limitations to better express this, as shown below.

Discussion, page 16, lines 314-318 (underlined text is new):

The results from our study should be interpreted in light of the following limitations. First, our findings only represent a snapshot of individuals’ hemodynamic status, preventing us from assessing longitudinal clinical outcomes and pathological adaptations that may occur with long-term exposure to a particular hypertension phenotype, including structural and physiological changes in heart and vessels.

Comment 8: The clinical implications of the current study should be clear and practical.

Regards

Response: Thank you. We edited the clinical implications paragraph to made it clearer, as below.

Discussion section, pages 15 and 16, lines 299-311 (underlined text is new or edited):

Our study also has important implications for personalizing the care of patients presenting to an outpatient clinic with elevated blood pressure. As hypertension remains as one of the biggest public health challenges worldwide,[1] there is urgency in determining possible ways of improving its treatment efficacy by using personalized therapies tailored to each patient’s characteristics. Although we and other studies have shown that, on average, there are significant hemodynamic differences by sex, we also found that there is substantial same-sex heterogeneity. Notably, as hypertension prevalence and arterial stiffening increase among women after menopause,[2-4] the distribution overlap of CI and SVRI among those older than 50 years was almost complete between women and men. Thus, our study indicates that sex alone is not a good proxy of the underlying hemodynamic patterns of patients with elevated blood pressure and should not be used clinically to assume the hypertension phenotype. Instead, it is necessary to measure these parameters directly if information about the hemodynamic profile is considered necessary to guide the antihypertensive therapy. 

References:

1. Global, regional, and national comparative risk assessment of 84 behavioural, environmental and occupational, and metabolic risks or clusters of risks for 195 countries and territories, 1990-2017: a systematic analysis for the Global Burden of Disease Study 2017. Lancet (London, England). 2018;392(10159):1923-94.

2. Cheng S, Xanthakis V, Sullivan LM, Vasan RS. Blood pressure tracking over the adult life course: patterns and correlates in the Framingham heart study. Hypertension. 2012;60(6):1393-9.

3. Zhou Y, Zhou X, Guo X, Sun G, Li Z, Zheng L, et al. Prevalence and risk factors of hypertension among pre- and post-menopausal women: a cross-sectional study in a rural area of northeast China. Maturitas. 2015;80(3):282-7. 

4. Lu Y, Pechlaner R, Cai J, Yuan H, Huang Z, Yang G, et al. Trajectories of age-related arterial stiffness in Chinese men and women. Journal of the American College of Cardiology. 2020;75(8):870-80.

REVIEWER 2

Comment 1: The cofounders missing some important factors as presence of diabetes, the types of hypertension ,the risk factors as dyslipidemia and the existence of chronic kidney disease. All these factors can contribute in risk stratification and further management of hypertension.

Response: Our aim was to describe sex differences in a hemodynamic snapshot of people with elevated blood pressure who visited an outpatient setting, regardless of their current management or chronic conditions. This is a real-world, pragmatic study showing that among a large outpatient clinic, there is marked heterogeneity in profiles and that sex is significantly associated with those profiles—but that there is a lot of overlap of the profiles of men and women.

We added the lack of comorbidities to the study’s limitations, as shown below.

Limitations, page 16, lines 319-324 (underlined text is new):

Second, although we had highly detailed hemodynamic information about each individual, the sociodemographic and clinical information (including comorbidities that may affect hemodynamic status) available for our analyses was limited. Of great importance is the lack of information regarding current antihypertensive medication usage, which could alter the hemodynamic phenotype and inclusion criteria (e.g., beta-blockers lowering CO or a patient with controlled hypertension not being included).

Comment 2: the age 50 is not clear as not standard for old age as WHO recommendations.

Response: We used 50 years not because it represents the age at which people are consider elder, but because it is the mean age of menopause among women in China.[1-3] See below 

Methods section, page 5, lines 96-103 (relevant text is underlined):

Variable definitions

We described demographic characteristics and hemodynamic parameters of blood pressure in women and men overall and by age. Considering that the mean age of natural menopause in China is reported as approximately 50 years of age, we stratified our study population as <50 years old and ≥50 years old. We used the World Health Organization recommended cutoff values for BMI classification in Asian populations, defining underweight as <18.5 kg/m2, normal weight from 18.5 kg/m2 to <23 kg/m2, overweight from 23 kg/m2 to <27.5 kg/m2, and obesity as ≥27.5 kg/m2. 

To make it clearer, we mention it in the introduction as shown below. 

Introduction section, pages 2 and 3, lines 41-48 (underlined text is new):

Accordingly, we used data from tens of thousands of individuals with elevated blood pressure from an outpatient setting in China to evaluate the overall patterns of sex differences in hemodynamic variables and to determine how these sex hemodynamic differences may vary with age. We also aimed to evaluate the distribution of these variables among women and men, and to what extent they overlap by sex. Furthermore, we stratified our analysis at the mean age of menopause in China because of its potential association with hemodynamic changes. Results from this study can advance our understanding of the association of sex with hemodynamic patterns in people with hypertension and suggest if sex could be used to guide therapy. 

References:

1. Dorjgochoo T, Kallianpur A, Gao YT, Cai H, Yang G, Li H, et al. Dietary and lifestyle predictors of age at natural menopause and reproductive span in the Shanghai Women's Health Study. Menopause. 2008;15(5):924-33. 

2. Wang M, Gong WW, Hu RY, Wang H, Guo Y, Bian Z, et al. Age at natural menopause and associated factors in adult women: Findings from the China Kadoorie Biobank study in Zhejiang rural area. PLoS One. 2018;13(4):e0195658. 

3. Song L, Shen L, Li H, Liu B, Zheng X, Zhang L, et al. Age at natural menopause and hypertension among middle-aged and older Chinese women. J Hypertens. 2018;36(3):594-600.

Comment 3: in table 1 what is meant by predominantly cardiac

Response: As stated in the footnote of Table 1 (found at the end of this document) and in the methods section, we defined a predominantly cardiac phenotype as high cardiac index with low or normal systemic vascular resistance index. This definition is consistent with previous research.[1-3]

Methods section, page 5, lines 96-107 (relevant text is underlined):

Variable definitions

We described demographic characteristics and hemodynamic parameters of blood pressure in women and men overall and by age. Considering that the mean age of natural menopause in China is reported as approximately 50 years of age, we stratified our study population as <50 years old and ≥50 years old. We used the World Health Organization recommended cutoff values for BMI classification in Asian populations, defining underweight as <18.5 kg/m2, normal weight from 18.5 kg/m2 to <23 kg/m2, overweight from 23 kg/m2 to <27.5 kg/m2, and obesity as ≥27.5 kg/m2. We defined a predominantly vascular hypertension phenotype as high SVRI (>2400 dynes·sec·cm-5·m2) with a low or normal CI (<2.5 L/min/m2 or 2.5–4 L/min/m2, respectively), and predominantly cardiac hypertension phenotype as high CI (>4 L/min/m2) with low or normal SVRI (<2000 dynes·sec·cm-5·m2 or 2000–2400 dynes·sec·cm-5·m2, respectively).

References:

1. Lu Y, Wang L, Wang H, Gu J, Ma ZJ, Lian Z, et al. Effectiveness of an impedance cardiography guided treatment strategy to improve blood pressure control in a real-world setting: results from a pragmatic clinical trial. Open Heart. 2021;8(2): e001719.

2. Mahajan S, Gu J, Caraballo C, Lu Y, Spatz ES, Zhao H, et al. Relationship of age with the hemodynamic parameters in individuals with elevated blood pressure. Journal of the American Geriatrics Society. 2020;68(7):1520-1528.

3. Mahajan S, Gu J, Lu Y, Khera R, Spatz ES, Zhang M, et al. Hemodynamic phenotypes of hypertension based on cardiac output and systemic vascular resistance. The American Journal of Medicine. 2020;133(4):e127-e139. 

Additional comment from the authors: please note that, for conciseness and consistency in the data reporting across text and tables, we edited the P values to express P values to 2 digits to the right of the decimal point, or to 3 digits if <0.01. During our thorough revision we also detected that a few P values in Table 1, Table S1, and Table S2 needed to be updated because of initial entry error. We apologize for the initial typographical errors, which have been corrected and tracked, and we confirm that none of the changes affect our findings or conclusions.

Tables

Table 1. Sex differences in clinical and hemodynamic variables by age group among adults with elevated blood pressure.

 All < 50 years old ≥ 50 years old

 Women

N=15,888 Men

N= 29194 P value Women

N = 4,384 Men

N = 15,512 P value Women

N = 11,504 Men

N = 13,682 P value

Age (years) 54.5 (11.8) 48.0 (13.0) <0.001 39.3 (8.1) 38.0 (7.3) <0.001 60.2 (6.9) 59.5 (7.2) <0.001

BMI (kg/m2) 24.4 (3.5) 25.5 (3.2) <0.001 23.5 (3.7) 25.7 (3.4) <0.001 24.8 (3.3) 25.2 (3.0) <0.001

Obesity* 2733 (17.2%) 6858 (23.49%) <0.001 585 (13.34%) 4057 (26.15%) <0.001 2148 (18.67%) 2801 (20.47%) <0.001

Region <0.001 <0.001 <0.001

 East 5965 (37.54%) 13275 (45.47%) 1956 (20.86%) 7419 (79.14%) 4009 (40.64%) 5856 (59.36%) 

 North 3665 (23.07%) 4156 (14.24%) 779 (29.59%) 1854 (70.41%) 2886 (55.63%) 2302 (44.37%) 

 South 2737 (17.23%) 4317 (14.79%) 686 (22.66%) 2341 (77.34%) 2051 (50.93%) 1976 (49.07%) 

 Southwest 3521 (22.16%) 7446 (25.51%) 963 (19.81%) 3898 (80.19%) 2558 (41.89%) 3548 (58.11%) 

Blood pressure (mmHg) 

 Systolic 139.0 (15.7) 136.7 (13.8) <0.001 131.2 (13.2) 133.7 (12.2) <0.001 142.0 (15.6) 140.1 (14.8) <0.001

 Diastolic 82.6 (9.00) 85.6 (8.9) <0.001 83.3 (7.9) 85.3 (8.9) <0.001 82.4 (9.4) 86.01 (9.0) <0.001

Hypertension phenotype 

 Predominantly cardiac† 2559 (16.11%) 5185 (17.76%) <0.001 1382 (31.52%) 3495 (22.53%) <0.001 1177 (10.23%) 1690 (12.35%) <0.001

 Predominantly vascular‡ 9780 (61.56%) 16396 (56.16%) <0.001 1666 (38.00%) 7197 (46.40%) <0.001 8114 (70.53%) 9199 (67.23%) <0.001

Low/normal CI & low/normal SVRI 3531 (22.22)% 7548 (25.86%) <0.001 1,330 (30.34%) 4,790(30.88%) 0.50 2,201 (19.13%) 2,758 (20.16%) 0.04

 High CI & high SVRI 18 (0.11%) 65 (0.22%) 0.01 6 (0.14%) 30 (0.19%) 0.56 12 (0.10%) 35 (0.26%) 0.01

ICG parameters 

 Heart rate (bpm) 69.4 (11.4) 69.5 (11.3) 0.63 72.6 (11.9) 70.8 (11.1) <0.001 68.2 (10.9) 68.0 (11.3) 0.15

 Stroke volume (mL) 72.9 (18.6) 88.8 (21.5) <0.001 80.0 (18.8) 93.0 (21.6) <0.001 70.2 (17.7) 84.0 (20.5) <0.001

 CO (L/min) 5.0 (1.4) 6.1 (1.5) <0.001 5.8 (1.4) 6.5 (1.4) <0.001 4.7 (1.2) 5.6 (1.4) <0.001

 CI (L/min/m2) 3.2 (0.8) 3.3 (0.8) <0.001 3.6 (0.9) 3.5 (0.7) <0.001 3.0 (0.8) 3.2 (0.7) <0.001

 SVR (dynes·sec·cm-5) 1744 (523) 1433 (389) <0.001 1471 (411) 1315 (324) <0.001 1848 (524) 1565 (412) <0.001

 SVRI (dynes·sec·cm-5·m2) 2734.1 (809.9) 2596.3 (677.2) <0.001 2326.0 (658.0) 2435.1 (598.6) <0.001 2889.6 (808.3) 2779.2 (713.8) <0.001

Data are presented as mean (SD) for continuous variables and n (%) for categorical variables.

* Obesity was defined as BMI ≥27.5 kg/m2

† A predominantly cardiac hypertension phenotype was determined by high CI with low or normal SVRI

‡ Predominantly vascular hypertension phenotype was determined by low or normal CI with high SVRI

Abbreviations: SD= Standard Deviation, BMI= Body Mass Index, ICG= Impedance Cardiography, SVR= Systemic Vascular Resistance, SVRI= Systemic Vascular Resistance Index, CO= Cardiac Output, CI= Cardiac Index.

Table 2. Unadjusted and Sequentially-Adjusted Association of Female Sex with Cardiac Output, Cardiac Index, Systemic Vascular Resistance, and Systemic Vascular Resistance Index, Overall and by Age Categories.

Hemodynamic Variable

 Female Sex β Coefficient (95% CI)

 Unadjusted Model Adjusted Model 1* Adjusted Model 2†

Cardiac Output, (L/min)

 Overall -1.07 (-1.1, -1.04) -0.79 (-0.82, -0.77) -0.78 (-0.8, -0.75)

 <50 years old -0.73 (-0.78, -0.68) -0.67 (-0.71, -0.62) -0.59 (-0.64, -0.54)

 ≥50 years old -0.90 (-0.93, -0.87) -0.86 (-0.89, -0.83) -0.86 (-0.89, -0.83)

Cardiac Index, (L/min/m2)

 Overall -0.15 (-0.16, -0.13) -0.01 (-0.03, 0)‡ -0.08 (-0.09, -0.06)

 <50 years old 0.14 (0.12, 0.17) 0.19 (0.16, 0.21) 0.07 (0.04, 0.09)

 ≥50 years old -0.14 (-0.16, -0.12) -0.12 (-0.14, -0.1) -0.15 (-0.16, -0.13)

Systemic Vascular Resistance, (dynes·sec·cm-5) 

 Overall 312 (303, 320) 225 (216, 233) 230 (221, 238)

 <50 years old 156 (144, 167) 138 (127, 149) 140 (128, 151)

 ≥50 years old 282 (271, 294) 271 (260, 282) 274 (262, 285)

Systemic Vascular Resistance Index, (dynes·sec·cm-5·m2)

 Overall 137.8 (123.7, 151.8) 6.0 (-7.6, 19.7)§ 73.5 (60.2, 86.8)

 <50 years old -109.1 (-129.7, -88.6) -146.9 (-166.6, -127.3) -31.7 (-51.2, -12.2)

 ≥50 years old 110.5 (91.7, 129.3) 88.5 (69.9, 107.2) 120.4 (102.4, 138.5)

* Model 1 was adjusted for age and region.

† Model 2 was adjusted for age, region, and body mass index.

‡P value=0.16

§P value=0.39

All other P values <0.001

S2 Table. Sex Differences in Clinical and Hemodynamic Variables by Nearest Neighbor Propensity Score Matched Subgroups.

 All

1:1 Matching <50 years old

1:1 Matching ≥50 years old

1:1 Matching

 Women

N=15,888 Men

N= 15,888 P value Women

N = 4,384 Men

N = 4,384 P value Women

N = 11,504 Men

N = 11,504 P value

Age, years mean (SD) 54.46 (11.83) 53.45 (12.72) <0.001 39.3 (8.07) 39.27 (7.5) 0.84 60.24 (6.9) 59.68 (7.24) <0.001

BMI (kg/m2), mean (SD) 24.4 (3.49) 24.53 (2.95) <0.001 23.49 (3.72) 23.65 (3.3) 0.04 24.75 (3.33) 24.97 (2.94) <0.001

Obese (BMI ≥27.5 kg/m2), n(%) 2733 (17.2%) 2284 (14.38%) <0.001 585 (13.34%) 483 (11.02%) <0.001 2148 (18.67%) 2086 (18.13%) 0.30

Region, n (%) <0.001 0.26 <0.001

East 5965 (48.74%) 6273 (51.26%) 1956 (50.46%) 1920 (49.54%) 4009 (46.58%) 4597 (53.42%) 

North 3665 (51.84%) 3405 (48.16%) 779 (47.94%) 846 (52.06%) 2886 (55.98%) 2269 (44.02%) 

South 2737 (50.39%) 2695 (49.61%) 686 (49.67%) 695 (50.33%) 2051 (52.32%) 1869 (47.68%) 

South West 3521 (50.04%) 3515 (49.96%) 963 (51.06%) 923 (48.94%) 2558 (48.02%) 2769 (51.98%) 

Blood pressure in mmHg, mean (SD) 

Systolic 139.02 (15.71) 138 (13.98) <0.001 131.21 (13.19) 131.52 (11.06) 0.04 142 (15.57) 140.36 (14.57) <0.001

Diastolic 82.64 (9.00) 83.62 (8.39) <0.001 83.32 (7.85) 83.37 (8.05) 0.23 82.37 (9.38) 84.61 (8.32) <0.001

ICG parameters, mean (SD) 

CO (L/min) 5.01 (1.35) 5.87 (1.43) <0.001 5.75 (1.41) 6.39 (1.42) <0.001 4.73 (1.21) 5.63 (1.35) <0.001

CI (L/min/m2) 3.19 (0.84) 3.31 (0.78) <0.001 3.64 (0.88) 3.59 (0.77) 0.013 3.02 (0.76) 3.18 (0.74) <0.001

SVR

(dynes·sec·cm-5) 1743.94 (523.42) 1475.6 (406.58) <0.001 1470.91 (411.34) 1306.85 (315.19) <0.001 1847.99 (524.09) 1552.76 (410.64) <0.001

SVRI

(dynes·sec·cm-5·m2) 2734.1 (809.92) 2606.44 (691.7) <0.001 2325.95 (658.03) 2322.25 (556.02) 0.78 2889.64 (808.27) 2741.28 (702.93) <0.001

SD= Standard Deviation, BMI= Body Mass Index, ICG= Impedance Cardiography, SVR= Systemic Vascular Resistance, SVRI= Systemic Vascular Resistance Index, CO= Cardiac Output, CI= Cardiac Index. 

Propensity Score Generation Model: Gender ~ Age + BMI + SBP + DBP + Region

---

## [Decision Letter · Decision Letter 1]

12 May 2022

PONE-D-21-16564R1

Hemodynamic Differences Between Women and Men with Elevated Blood Pressure in China: A Non-Invasive Assessment of 45,082 Adults Using Impedance Cardiography

PLOS ONE

Dear Dr. Krumholz,

Thank you for submitting your manuscript to PLOS ONE. After careful consideration, we feel that it has merit but does not fully meet PLOS ONE’s publication criteria as it currently stands. Therefore, we invite you to submit a revised version of the manuscript that addresses the points raised during the review process.

The authors improved the manuscript but appear to have ignored comments of reviewer 2. If they respond sufficiently to these comments the manuscript can be accepted. Please check carefully once more the initial comments of reviewer 2 as listed below:

- The cofounders missing some important factors as presence of diabetes , the types of hypertension ,the risk factors as dyslipidemia and the existence of chronic kidney disease .all these factors can contribute in risk stratification and further management of hypertension.

- the age 50 is not clear as not standard for old age as WHO recommendations.

- in table 1 what is meant by predominantly cardiac

We look forward to receiving your revised manuscript.

Kind regards,

Johannes Vogel

Academic Editor

PLOS ONE

Journal Requirements:

Reviewers' comments:

Reviewer's Responses to Questions

**Comments to the Author**

1. If the authors have adequately addressed your comments raised in a previous round of review and you feel that this manuscript is now acceptable for publication, you may indicate that here to bypass the “Comments to the Author” section, enter your conflict of interest statement in the “Confidential to Editor” section, and submit your "Accept" recommendation.

Reviewer #1: (No Response)

Reviewer #2: (No Response)

2. Is the manuscript technically sound, and do the data support the conclusions?

Reviewer #1: Yes

Reviewer #2: Partly

3. Has the statistical analysis been performed appropriately and rigorously? 

Reviewer #1: Yes

Reviewer #2: N/A

4. Have the authors made all data underlying the findings in their manuscript fully available?

Reviewer #1: No

Reviewer #2: No

5. Is the manuscript presented in an intelligible fashion and written in standard English?

Reviewer #1: Yes

Reviewer #2: No

6. Review Comments to the Author

Reviewer #1: The manuscript has been improved. However, all the authors responses to our comments should be highlighted in the text. All the numbers across the manuscript should be revised carefully for correctness and coherence.

Regards

Reviewer #2: i rejected the manuscript titled "Hemodynamic Differences Between Women and Men with Elevated Blood Pressure in

China: A Non-Invasive Assessment of 45,082 Adults Using Impedance Cardiography" as no reasonable response for reviewers comments.

7. PLOS authors have the option to publish the peer review history of their article (what does this mean?). If published, this will include your full peer review and any attached files.

Reviewer #1: **Yes: **Rami Riziq Yousef Abumuaileq

Reviewer #2: No

---

## [Author Response · Author response to Decision Letter 1]

20 May 2022

Please see the attached document for our responses in appropriate format. They are pasted also below.

Response To Comments

PONE-D-21-16564: Hemodynamic Differences Between Women and Men with Elevated Blood Pressure in China: A non-invasive assessment of 45,082 adults using impedance cardiography

We have responded to each comment (reproduced in bold) and detailed our changes to the manuscript. In this document, quoted text is presented indented and new additions (or relevant text when specified) underlined. The page and line numbers that we reference below are based on the clean version of our revised manuscript. New references are listed at the end of each response. 

ACADEMIC EDITOR

Comment 1: Thank you for submitting your manuscript to PLOS ONE. After careful consideration, we feel that it has merit but does not fully meet PLOS ONE’s publication criteria as it currently stands. Therefore, we invite you to submit a revised version of the manuscript that addresses the points raised during the review process.

The authors improved the manuscript but appear to have ignored comments of reviewer 2. If they respond sufficiently to these comments the manuscript can be accepted. Please check carefully once more the initial comments of reviewer 2 as listed below:

Response: We appreciate the academic editor comments and the opportunity to revise our work. This may be a misunderstanding. We did respond to the Reviewer 2 in our initial response letter and made some modifications to the manuscript accordingly. We apologize if our responses were somehow left out from the document that you reviewed. We reviewed each comment once more and included our responses below. Please note that we highlighted these changes in the attached tracked version of the manuscript.

Comment 2: 

- The cofounders missing some important factors as presence of diabetes, the types of hypertension ,the risk factors as dyslipidemia and the existence of chronic kidney disease. All these factors can contribute in risk stratification and further management of hypertension.

Response: Our aim was to describe sex differences in a hemodynamic snapshot of people with elevated blood pressure who visited an outpatient setting, regardless of their current management or chronic conditions. This is a real-world, pragmatic study showing that among a large outpatient clinic, there is marked heterogeneity in profiles and that sex is significantly associated with those profiles—but that there is a lot of overlap of the profiles of men and women.

We added the lack of comorbidities to the study’s limitations, as shown below.

Limitations, page 16, lines 321-326 (underlined text is new):

Second, although we had highly detailed hemodynamic information about each individual, the sociodemographic and clinical information (including comorbidities that may affect hemodynamic status) available for our analyses was limited. Of great importance is the lack of information regarding current antihypertensive medication usage, which could alter the hemodynamic phenotype and inclusion criteria (e.g., beta-blockers lowering CO or a patient with controlled hypertension not being included).

- Comment 3: the age 50 is not clear as not standard for old age as WHO recommendations.

Response: We used 50 years not because it represents the age at which people are consider elder, but because it is the mean age of menopause among women in China.[1-3] See below 

Methods section, page 5, lines 98-105 (relevant text is underlined):

Variable definitions

We described demographic characteristics and hemodynamic parameters of blood pressure in women and men overall and by age. Considering that the mean age of natural menopause in China is reported as approximately 50 years of age, we stratified our study population as <50 years old and ≥50 years old. We used the World Health Organization recommended cutoff values for BMI classification in Asian populations, defining underweight as <18.5 kg/m2, normal weight from 18.5 kg/m2 to <23 kg/m2, overweight from 23 kg/m2 to <27.5 kg/m2, and obesity as ≥27.5 kg/m2. 

To make it clearer, we mention it in the introduction as shown below. 

Introduction section, pages 2 and 3, lines 41-48 (underlined text is new):

Accordingly, we used data from tens of thousands of individuals with elevated blood pressure from an outpatient setting in China to evaluate the overall patterns of sex differences in hemodynamic variables and to determine how these sex hemodynamic differences may vary with age. We also aimed to evaluate the distribution of these variables among women and men, and to what extent they overlap by sex. Furthermore, we stratified our analysis at the mean age of menopause in China because of its potential association with hemodynamic changes. Results from this study can advance our understanding of the association of sex with hemodynamic patterns in people with hypertension and suggest if sex could be used to guide therapy. 

References:

1. Dorjgochoo T, Kallianpur A, Gao YT, Cai H, Yang G, Li H, et al. Dietary and lifestyle predictors of age at natural menopause and reproductive span in the Shanghai Women's Health Study. Menopause. 2008;15(5):924-33. 

2. Wang M, Gong WW, Hu RY, Wang H, Guo Y, Bian Z, et al. Age at natural menopause and associated factors in adult women: Findings from the China Kadoorie Biobank study in Zhejiang rural area. PLoS One. 2018;13(4):e0195658. 

3. Song L, Shen L, Li H, Liu B, Zheng X, Zhang L, et al. Age at natural menopause and hypertension among middle-aged and older Chinese women. J Hypertens. 2018;36(3):594-600.

- Comment 4: in table 1 what is meant by predominantly cardiac

Response: As stated in the footnote of Table 1 (see underlined footnote in Table 1 below) and in the methods section, we defined a predominantly cardiac phenotype as high cardiac index with low or normal systemic vascular resistance index. This definition is consistent with previous research.[1-3]

Methods section, page 5, lines 98-109 (relevant text is underlined):

Variable definitions

We described demographic characteristics and hemodynamic parameters of blood pressure in women and men overall and by age. Considering that the mean age of natural menopause in China is reported as approximately 50 years of age, we stratified our study population as <50 years old and ≥50 years old. We used the World Health Organization recommended cutoff values for BMI classification in Asian populations, defining underweight as <18.5 kg/m2, normal weight from 18.5 kg/m2 to <23 kg/m2, overweight from 23 kg/m2 to <27.5 kg/m2, and obesity as ≥27.5 kg/m2. We defined a predominantly vascular hypertension phenotype as high SVRI (>2400 dynes·sec·cm-5·m2) with a low or normal CI (<2.5 L/min/m2 or 2.5–4 L/min/m2, respectively), and predominantly cardiac hypertension phenotype as high CI (>4 L/min/m2) with low or normal SVRI (<2000 dynes·sec·cm-5·m2 or 2000–2400 dynes·sec·cm-5·m2, respectively).

References:

1. Lu Y, Wang L, Wang H, Gu J, Ma ZJ, Lian Z, et al. Effectiveness of an impedance cardiography guided treatment strategy to improve blood pressure control in a real-world setting: results from a pragmatic clinical trial. Open Heart. 2021;8(2): e001719.

2. Mahajan S, Gu J, Caraballo C, Lu Y, Spatz ES, Zhao H, et al. Relationship of age with the hemodynamic parameters in individuals with elevated blood pressure. Journal of the American Geriatrics Society. 2020;68(7):1520-1528.

3. Mahajan S, Gu J, Lu Y, Khera R, Spatz ES, Zhang M, et al. Hemodynamic phenotypes of hypertension based on cardiac output and systemic vascular resistance. The American Journal of Medicine. 2020;133(4):e127-e139. 

Table 1. Sex differences in clinical and hemodynamic variables by age group among adults with elevated blood pressure.

 All < 50 years old ≥ 50 years old

 Women

N=15,888 Men

N= 29194 P value Women

N = 4,384 Men

N = 15,512 P value Women

N = 11,504 Men

N = 13,682 P value

Age (years) 54.5 (11.8) 48.0 (13.0) <0.001 39.3 (8.1) 38.0 (7.3) <0.001 60.2 (6.9) 59.5 (7.2) <0.001

BMI (kg/m2) 24.4 (3.5) 25.5 (3.2) <0.001 23.5 (3.7) 25.7 (3.4) <0.001 24.8 (3.3) 25.2 (3.0) <0.001

Obesity* 2733 (17.2%) 6858 (23.49%) <0.001 585 (13.34%) 4057 (26.15%) <0.001 2148 (18.67%) 2801 (20.47%) <0.001

Region <0.001 <0.001 <0.001

 East 5965 (37.54%) 13275 (45.47%) 1956 (20.86%) 7419 (79.14%) 4009 (40.64%) 5856 (59.36%) 

 North 3665 (23.07%) 4156 (14.24%) 779 (29.59%) 1854 (70.41%) 2886 (55.63%) 2302 (44.37%) 

 South 2737 (17.23%) 4317 (14.79%) 686 (22.66%) 2341 (77.34%) 2051 (50.93%) 1976 (49.07%) 

 Southwest 3521 (22.16%) 7446 (25.51%) 963 (19.81%) 3898 (80.19%) 2558 (41.89%) 3548 (58.11%) 

Blood pressure (mmHg) 

 Systolic 139.0 (15.7) 136.7 (13.8) <0.001 131.2 (13.2) 133.7 (12.2) <0.001 142.0 (15.6) 140.1 (14.8) <0.001

 Diastolic 82.6 (9.00) 85.6 (8.9) <0.001 83.3 (7.9) 85.3 (8.9) <0.001 82.4 (9.4) 86.01 (9.0) <0.001

Hypertension phenotype 

 Predominantly cardiac† 2559 (16.11%) 5185 (17.76%) <0.001 1382 (31.52%) 3495 (22.53%) <0.001 1177 (10.23%) 1690 (12.35%) <0.001

 Predominantly vascular‡ 9780 (61.56%) 16396 (56.16%) <0.001 1666 (38.00%) 7197 (46.40%) <0.001 8114 (70.53%) 9199 (67.23%) <0.001

Low/normal CI & low/normal SVRI 3531 (22.22)% 7548 (25.86%) <0.001 1,330 (30.34%) 4,790(30.88%) 0.50 2,201 (19.13%) 2,758 (20.16%) 0.04

 High CI & high SVRI 18 (0.11%) 65 (0.22%) 0.01 6 (0.14%) 30 (0.19%) 0.56 12 (0.10%) 35 (0.26%) 0.01

ICG parameters 

 Heart rate (bpm) 69.4 (11.4) 69.5 (11.3) 0.63 72.6 (11.9) 70.8 (11.1) <0.001 68.2 (10.9) 68.0 (11.3) 0.15

 Stroke volume (mL) 72.9 (18.6) 88.8 (21.5) <0.001 80.0 (18.8) 93.0 (21.6) <0.001 70.2 (17.7) 84.0 (20.5) <0.001

 CO (L/min) 5.0 (1.4) 6.1 (1.5) <0.001 5.8 (1.4) 6.5 (1.4) <0.001 4.7 (1.2) 5.6 (1.4) <0.001

 CI (L/min/m2) 3.2 (0.8) 3.3 (0.8) <0.001 3.6 (0.9) 3.5 (0.7) <0.001 3.0 (0.8) 3.2 (0.7) <0.001

 SVR (dynes·sec·cm-5) 1744 (523) 1433 (389) <0.001 1471 (411) 1315 (324) <0.001 1848 (524) 1565 (412) <0.001

 SVRI (dynes·sec·cm-5·m2) 2734.1 (809.9) 2596.3 (677.2) <0.001 2326.0 (658.0) 2435.1 (598.6) <0.001 2889.6 (808.3) 2779.2 (713.8) <0.001

Data are presented as mean (SD) for continuous variables and n (%) for categorical variables.

* Obesity was defined as BMI ≥27.5 kg/m2

† A predominantly cardiac hypertension phenotype was determined by high CI with low or normal SVRI

‡ Predominantly vascular hypertension phenotype was determined by low or normal CI with high SVRI

Abbreviations: SD= Standard Deviation, BMI= Body Mass Index, ICG= Impedance Cardiography, SVR= Systemic Vascular Resistance, SVRI= Systemic Vascular Resistance Index, CO= Cardiac Output, CI= Cardiac Index.

---

## [Editor Report · Decision Letter 2]

31 May 2022

Hemodynamic Differences Between Women and Men with Elevated Blood Pressure in China: A Non-Invasive Assessment of 45,082 Adults Using Impedance Cardiography

PONE-D-21-16564R2

Dear Dr. Krumholz,

We’re pleased to inform you that your manuscript has been judged scientifically suitable for publication and will be formally accepted for publication once it meets all outstanding technical requirements.

Kind regards,

Johannes Vogel

Academic Editor

PLOS ONE
---

## [Editor Report · Acceptance letter]

3 Jun 2022

PONE-D-21-16564R2 

Hemodynamic Differences Between Women and Men with Elevated Blood Pressure in China: A non-invasive assessment of 45,082 adults using impedance cardiography 

Dear Dr. Krumholz:

I'm pleased to inform you that your manuscript has been deemed suitable for publication in PLOS ONE. Congratulations! Your manuscript is now with our production department. 

Kind regards, 

on behalf of

Professor Johannes Vogel 

Academic Editor

PLOS ONE